# New calculation model and application research on weak water-flushed zones distribution prediction in radial flow well patterns of heterogeneous oil reservoirs

**Mengyun Li, Guicai Zhang** *, **Haihua Pei, Ping Jiang, Liu Yang**

College of Petroleum Engineering, China University of Petroleum, Qingdao, China

* zhanggc@upc.edu.cn

## Abstract

After prolonged waterflooding development, the main integrated oil reservoirs in the X Oilfield have largely entered the late stage of waterflooding, with an average water cut reaching approximately 98%. However, a significant amount of remaining oil still accumulates in the weak water-flushed zones of low-permeability layers, which presents substantial development potential. Therefore, accurately predicting the distribution of weak water-flushed zones is crucial for optimizing extraction measures and enhancing recovery rates. However, during the development process, especially in the ultra-high water cut stage, the continuous changes in oil-water flow resistance in multi-layer heterogeneous reservoirs make it very difficult to accurately predict the distribution of weak water-flushed zones. At present, the prediction of weak water-flushed zone is highly dependent on complex physical models, the computational process is cumbersome, and the relevant research on the distribution of weak water-flushed zone under radial flow is scarce. To address this, based on Darcy's Law and Buckley-Leverett's non-piston water flooding theory, taking the flow resistance coefficient as the index, this paper deduces the water drive front advancement equation for different permeability layers under radial flow conditions in heterogeneous reservoirs (including barriers). This study established an iterative calculation method for the advancement distance of the water drive front in low-permeability layers, forming a simple and convenient method for describing the distribution of weak water-flushed zones in heterogeneous reservoirs. On this basis, the accuracy of the proposed weak water-flushed zone prediction method is verified through numerical simulation experiments based on the geological model of the X oil field. The results show that the relative error between the calculated results of new calculation model and numerical simulation results is less than 5%, indicating that the method is simple to use and highly accurate. Finally, using the established prediction method, we calculated the limiting conditions for the existence of weak water-flushed zones in the X oil field

**Data availability statement:** All relevant data are within the paper and its Supporting Information files.

**Funding:** The author(s) received no specific funding for this work.

**Competing interests:** The authors have declared that no competing interests exist.

test area. Under the oil field's model conditions, when the permeability ratio exceeds 2.4, weak water-flushed zones exist in the low-permeability layers when the water cut reaches 98%. Furthermore, the initiation limit of the weak water-flushed zone is clarified: under the model conditions (with a permeability ratio of 10), the weak water-flushed zone in the low-permeability layers can be initiated when the permeability ratio decreases to 2.4 after plugging adjustments or the viscosity of the injected displacing medium reaches 10 mPa·s. This prediction method can provide technical guidance for optimizing decision-making in the efficient development of ultra-high water cut oil fields.

## 1. Introduction

In the waterflooding stage of multi-layer heterogeneous reservoirs, a five-spot well pattern is commonly used for development. Due to cost considerations, the same development system is typically applied across layers in the vertical direction. As the reservoir enters a high-water-cut stage, interlayer conflicts become more pronounced, and the variation in water absorption among layers becomes significant. This results in lower development efficiency in low-permeability layers, with weak water-flushed zones being widespread and difficult to predict [1–4]. Accurately determining the position of the waterflood front in low-permeability zones is crucial, as it helps in selecting appropriate measures and parameters for subsequent production. This is of great importance in enhancing the recovery factor of multi-layer heterogeneous reservoirs [5–6].

Currently, there are few studies on the methods for determining the specific position of the waterflood front in heterogeneous layers at different water cut stages [7–9]. Early research mainly relied on the piston-like waterflooding theory for related derivations. Scholars such as Osman and Dyes studied the development dynamics of heterogeneous reservoirs under piston-type waterflooding conditions and the development of the reservoir after the water breakthrough. However, in actual field operations, the waterflooding process is carried out in a non-piston displacement mode, and there is a significant discrepancy between the conclusions derived from piston-type waterflooding theory and the actual field conditions [10–11]. Later, some scholars, based on the non-piston flooding theory, established models and made predictions in a one-dimensional direction. Scholars like Feng Qihong and Zhou Yingfang, through theoretical derivations and experimental validation, proposed new methods for calculating various development indicators for heterogeneous reservoirs under one-dimensional linear flow conditions using non-piston waterflooding [12–13]. Subsequently, Sun Qiang and others expanded the scope of flow state studies, considering radial flow conditions, and conducted research on predicting development indicators for heterogeneous reservoirs. They provided production parameters, such as liquid production rates at different times. However, these studies still failed to accurately determine the specific position of the waterflood front in the subsurface reservoir and could not reliably predict the distribution of remaining oil [14]. Based on existing field-based studies, it can be observed that most reservoirs in China primarily

adopt five-spot or inverted nine-spot well patterns. The overall waterflooding flow in these reservoirs typically follows a radial flow pattern. Therefore, combining field data with research on the current displacement flow patterns and development states to analyze the distribution of remaining oil has greater potential for practical application in subsequent development [15–16]. Despite previous research efforts, three primary challenges remain: 1. Some methods rely heavily on complex physical models, increasing computational workload in practical applications. 2. Existing calculation processes are cumbersome, involving large computational demands. 3.Methods derived under linear flow conditions do not align well with the actual flow characteristics and well pattern distributions observed in oilfields [17–21].

Given the current research progress and development needs, this study uses the five-spot well pattern as the development well network. It applies the two-phase oil-water flow theory—Buckley-Leverett waterflooding theory—and combines Darcy's law to deduce the flow resistance of each permeability band. Iterative calculations are performed to determine the distribution area of weak water-flushed zones under radial flow conditions in multilayer heterogeneous reservoirs. This method provides important guidance for accurately determining the overall remaining oil distribution in heterogeneous formations and further improving the recovery factor of waterflooded reservoirs [22–24].

## 2. Experimental methodology

In early research on the waterflooding characteristics of reservoirs during the ultra-high water cut stage, reservoirs in consolidated oilfields were classified into three distinct waterflooding zones based on the remaining oil saturation: extreme water-washing zone, strong water-flooded zone, and weak water-flushed zone. The areas of the reservoir where the oil saturation exceeds the oil saturation at the waterflood front are designated as weak water-flushed zones [25]. Research indicates that weak water-flushed zones are the primary regions where remaining oil accumulates in waterflooded reservoirs. Accurately predicting the distribution of weak water-flushed zones in heterogeneous reservoirs is crucial for further improving recovery rates.

Given the limitations of earlier studies, which were often complex, time-consuming, and lacked practical applicability, this study focuses on developing a reservoir model and deriving a new, simplified, and practical method for predicting the distribution of weak water-flushed zones. This new method aims to overcome the shortcomings of previous approaches and provide a more effective tool for enhancing the recovery factor in heterogeneous reservoirs [26–30]. And the accuracy of this method is validated through numerical simulation tests, demonstrating its reliability and practical feasibility [31].

### 2.1. Prediction and calculation method for interlayer weak water-flushed zone distribution

Considering the heterogeneous characteristics of reservoirs with significant permeability differences between high- and low-permeability formations, this study investigates the radial flow-driven oil displacement process in a five-spot well pattern.

In waterflooding development, if the injected water exerts a strong influence, the elastic effects of the fluid and rock can generally be neglected. When the external pressure differential is substantial (e.g., during a stable waterflooding process), water mainly infiltrates the oil zone due to the external pressure difference. Under such conditions, capillary forces exert a relatively weak influence on the overall reservoir, while gravity segregation becomes more significant when there is a large density difference between oil and water and when the oil layer is thick. However, for the heterogeneous reservoir under discussion—where interlayer weak water-flushed zones exist due to the presence of impermeable barriers—the impact of density differences on the state of the oil-water mixing zone is relatively minor. Consequently, the most significant factor influencing the distribution of weak water-flushed zones in the reservoir is the permeability ratio between high- and low-permeability formations.

Based on the above analysis of influencing factors, the following assumptions are made for the non-piston-like waterflooding displacement process:

(1) There is a significant permeability ratio between high- permeability and low-permeability layers, while the oil layer thickness, flow cross-section area, and injection-production well placement remain identical;

(2) Both oil phase and water phase follow Darcy's law under radial flow conditions, with the displacement occurring in a non-piston-like manner;

(3) The effects of gravity, capillary pressure, gravitational segregation, and startup pressure gradients are neglected. Both the rock and the fluids are assumed to be incompressible;

(4) The reservoir operates under commingled injection and production with injection-production balance, interlayer crossflow is ignored, and the water injection in each layer is allocated according to the respective flow resistance of each layer.

A schematic representation of the reservoir and well pattern is shown in Fig 1.
Based on Darcy's Law:

$$Q = vA = 2\pi rh\frac{K}{\mu} \times \frac{dp}{dr} \tag{1}$$

After separating the variables and integrating, we obtain:

$$\int_{p_w}^{p_e} dp = \frac{Q\mu}{2\pi Kh} \int_{r_w}^{r_e} \frac{dr}{r} \tag{2}$$

The production rate for radial flow under single-phase flow conditions is given by:

$$Q = \frac{2\pi Kh(p_e - p_w)}{\mu \frac{r_e}{r_w}} \tag{3}$$

Based on the simplified form of Darcy's Law:

$$Q = \frac{\Delta P}{R} \tag{4}$$

The radial flow seepage resistance can be obtained:

$$R = \frac{\mu \frac{r_e}{r_w}}{2\pi Kh} \tag{5}$$

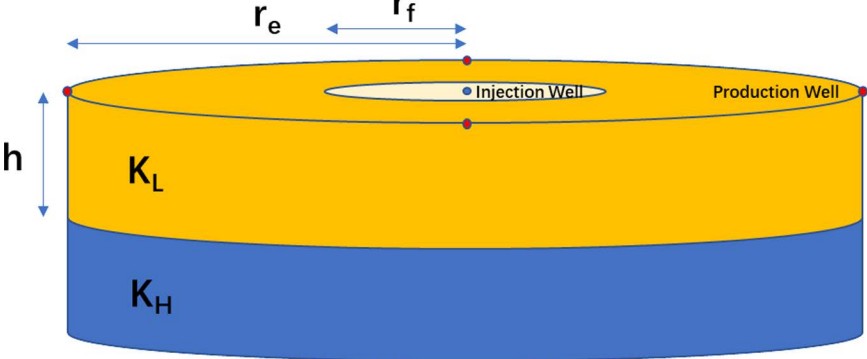

**Fig 1. Low-high permeability two-layer heterogeneous reservoir schematic drawing.**

For the waterflooding process in a dual-layer heterogeneous formation, different oil-water distribution states and relational patterns exist before and after the waterflood front breaks through the production well in the high-permeability layer. Therefore, in the subsequent derivation and analysis, the stages before and after the waterflood front breaks through the production well are derived and calculated separately.

According to the preset conditions of the waterflooding model, the injected water is allocated between the high and low permeability layers based on their respective flow resistances. Consequently, the flow resistance in the reservoir is analyzed separately for the stages before and after the waterflood front breaks through the production well as follows:

Before the Waterflood Front Breaks Through the Production Well

In the region where $r < R_f$, which is the two-phase flow area of oil and water, the seepage resistance of the water phase is considered

$$R_w = \frac{\mu_w \ln \frac{r_f}{r_w}}{2\pi K h K_{rw}(\overline{S_w})}$$

(6)

The oil phase seepage resistance is:

$$R_o = \frac{\mu_o \ln \frac{r_f}{r_w}}{2\pi K h K_{ro}(\overline{S_w})}$$

(7)

Based on $Q = Q_w + Q_o$ and Equation 4, $\frac{1}{R} = \frac{1}{R_o} + \frac{1}{R_w}$, the total seepage resistance in the two-phase oil-water region is obtained as follows:

$$R_{wo} = \frac{R_w R_o}{R_w + R_o} = \frac{\ln \frac{r_f}{r_w}}{2\pi K h} \frac{1}{\frac{K_{rw}(\overline{S_w})}{\mu_w} + \frac{K_{ro}(\overline{S_w})}{\mu_o}}$$

(8)

The resistance in the single-phase oil flow region is:

$$R_o = \frac{\mu_o \ln \frac{r_e}{r_f}}{2\pi K h K_{ro}(S_{wc})}$$

(9)

Thus, the total seepage resistance is:

$$R = R_{wo} + R_o = \frac{\ln \frac{r_f}{r_w}}{2\pi K h} \frac{1}{\frac{K_{rw}(\overline{S_w})}{\mu_w} + \frac{K_{ro}(\overline{S_w})}{\mu_o}} + \frac{\mu_o \ln \frac{r_e}{r_f}}{2\pi K h K_{ro}(S_{wc})}$$

(10)

(1) After the waterflood front breaks through at the production well, the seepage resistance for the two-phase oil-water flow is:

$$R_{wo} = \frac{R_w R_o}{R_w + R_o} = \frac{\ln \frac{r_e}{r_w}}{2\pi K h} \frac{1}{\frac{K_{rw}(\overline{S_w})}{\mu_w} + \frac{K_{ro}(\overline{S_w})}{\mu_o}}$$

(11)

Based on the above method for calculating seepage resistance coefficients, the seepage resistance in both high-permeability and low-permeability layers is analyzed for when the waterflood front breaks through the production well and after the breakthrough. The advancement radius of the waterflood front in the low-permeability layer is then calculated based on the ratio of seepage resistance between the high and low-permeability layers.

During multilayer commingled injection and production in reservoirs with varying permeability, the pressure differential between the injection and production ends remains equal, and the water injection split ratio is inversely proportional to the

ratio of seepage resistance between different layers. However, in the water flooding process, as the water front advances, the seepage resistance of the reservoir decreases continuously. Due to the disparity in flow distribution between high- and low-permeability layers, the ratio of their seepage resistance also varies dynamically. Therefore, analyzing the water flooding process solely based on the initial seepage resistance state of the reservoir may lead to significant errors.

Since high-permeability reservoirs have a relatively low initial seepage resistance and receive a greater proportion of injected water as water flooding progresses, the water front in high-permeability layers advances more rapidly, resulting in a higher rate of seepage resistance reduction. Consequently, a qualitative analysis suggests that the ratio of the water front advancement distance between low- and high-permeability layers is significantly lower than the permeability ratio of low- to high-permeability layers. By employing an iterative computation approach, the discrepancy between the calculated and actual interlayer seepage resistance ratio can be minimized, thereby enabling a quantitative determination of the advancement distance of the water drive front in the low-permeability layer.

(1) When the Waterflood Front Breaks Through the Production Well:

$$S_L^{max} = \frac{K_L}{K_H} \cdot S$$

(12)

From the formula for the area of a circle:

$$r_{fL}^{max} = \sqrt{\frac{K_L}{K_H}} \cdot r_e$$

(13)

The minimum total resistance in the low-permeability layer is given by:

$$R_L^{min} = \frac{\ln \frac{r_{fL}^{max}}{r_w}}{2\pi K_L h} \frac{1}{\frac{K_{rw}(\overline{S_w})}{\mu_w} + \frac{K_{ro}(\overline{S_w})}{\mu_o}} + \frac{\mu_o \ln \frac{r_e}{r_{fL}^{max}}}{2\pi K_L h K_{ro}(S_{wc})}$$

$$= \frac{\ln \frac{\sqrt{\frac{K_L}{K_H}} \cdot r_e}{r_w}}{2\pi K_L h} \frac{1}{\frac{K_{rw}(\overline{S_w^{Lwo}})}{\mu_w} + \frac{K_{ro}(\overline{S_w^{Lwo}})}{\mu_o}} + \frac{\mu_o \ln \frac{r_e}{\sqrt{\frac{K_L}{K_H}} \cdot r_e}}{2\pi K_L h K_{ro}(S_{wc})}$$

$$= \frac{\ln \frac{\frac{K_L}{K_H} \cdot r_e^2}{r_w^2}}{4\pi K_L h} \frac{1}{\frac{K_{rw}(\overline{S_w^{Lwo}})}{\mu_w} + \frac{K_{ro}(\overline{S_w^{Lwo}})}{\mu_o}} + \frac{\mu_o \ln \frac{K_L}{K_H}}{4\pi K_L h K_{ro}(S_{wc})}$$

$$= \frac{1}{4\pi K_L h} \left[ \frac{\ln \frac{\frac{K_L}{K_H} \cdot r_e^2}{r_w^2}}{\frac{K_{rw}(\overline{S_w^{Lwo}})}{\mu_w} + \frac{K_{ro}(\overline{S_w^{Lwo}})}{\mu_o}} + \frac{\mu_o \ln \frac{K_L}{K_H}}{K_{ro}(S_{wc})} \right]$$

(14)

The seepage resistance in the high-permeability layer is:

$$R_H = R_{wo} = \frac{R_w R_o}{R_w + R_o} = \frac{\ln \frac{r_e}{r_w}}{2\pi K_H h} \frac{1}{\frac{K_{rw}(\overline{S_w^H})}{\mu_w} + \frac{K_{ro}(\overline{S_w^H})}{\mu_o}}$$

(15)

Iterative calculations are performed for the seepage resistance $R_L^{min}$ in the low-permeability layer and the advancement radius $r_f$ of the waterflood front:

The initial seepage resistance ratio between the low-permeability and high-permeability layers is:

$$C_R^1 = \frac{R_L^{min}}{R_H}$$

(16)

The initial advancing area of the waterflood front in the low-permeability layer for the iterative process is:

$$S_{fL}^1 = \frac{S_e}{C_R^1} = \pi r_e^2 \frac{R_H}{R_L^{min}}$$

(17)

The initial advancing radius of the waterflood front for the iterative process is:

$$r_{fL}^1 = r_e \sqrt{\frac{R_H}{R_L^{min}}}$$

(18)

The seepage resistance in the low-permeability layer is:

$$R_L^1 = \frac{\ln \frac{r_{fL}^1}{r_w}}{2\pi K_L h} \frac{1}{\frac{K_{rw}(S_w^{Lwo})}{\mu_w} + \frac{K_{ro}(S_w^{Lwo})}{\mu_o}} + \frac{\mu_o \ln \frac{r_e}{r_{fL}^1}}{2\pi K_L h K_{ro}(S_{wc})}$$

(19)

The initial seepage resistance ratio between the low-permeability and high-permeability layers is:

$$C_R^2 = \frac{R_L^1}{R_H}$$

(20)

The iterative advancement area of the waterflood front in the low-permeability layer $S_{fL}^2$:

$$S_{fL}^2 = \frac{S_e}{C_R^2} = \pi r_e^2 \frac{R_H}{R_L^1}$$

(21)

Thus, the iterative advancement radius of the waterflood front $r_{fL}^2$ is:

$$r_{fL}^2 = r_e \sqrt{\frac{R_H}{R_L^1}}$$

(22)

Repeat the iterative calculations until the relative error between two consecutive iterations is less than 0.1%. The calculated advancement radius of the waterflood front in the low-permeability layer $r_{fL}^i$ will be the advancement radius of the waterflood front in the low-permeability layer when the waterflood front in the high-permeability layer breaks through at the production well.

For reservoirs with more than two heterogeneous layers of varying permeability, since the pressure differential between the injection and production ends remains equal across all layers, the seepage resistance of each layer can be analogously calculated. By applying the same iterative computation method, the advance radius of the water drive front in lower-permeability formations can be determined.

(2) After the Waterflood Front Breaks Through the Production Well:

If the effects of capillary forces and gravity are neglected, the Leverett function (Equation 23), also known as the fractional flow equation, can be derived using Darcy's Law and the continuity equation.

$$f_w = \frac{\frac{K_{rw}}{\mu_w}}{\frac{K_{rw}}{\mu_w} + \frac{K_{ro}}{\mu_o}} = \left[ 1 + \frac{K_{ro}\mu_w}{K_{rw}\mu_o} \right]^{-1}$$

(23)

 

To analyze the state of the weak water-flushed zone at different water-cut stages, the water saturation parameter is introduced into the formula. A commonly used model for describing the quantitative relationship between the oil and water relative permeability ratio and the water saturation is the equation proposed by Craft et al., as shown in Equation (24).

$$\frac{K_{ro}}{K_{rw}} = ae^{-bS_w}$$
(24)

By combining Equations 23 and 24, we obtain:

$$\frac{1-f_w}{f_w} = \frac{a\mu_w}{\mu_o}e^{-bS_w}$$
(25)

Taking the logarithm of both sides of Equation 25 yields:

$$\ln\left(\frac{1-f_w}{f_w}\right) = \ln\left(\frac{a\mu_w}{\mu_o}\right) - bS_w$$
(26)

By fitting the values of a and b, the water saturation $S_{w2}$ at the production well, corresponding to the production well water cut $f_{w2}$, can be determined.

For a given high-permeability layer water cut $f_w(S_{w2})$ at the production well, the average water saturation $\overline{S_w^{f_w^t}}$ with that water cut at time $t$ in the high-permeability layer can be calculated:

$$\overline{S_w^{f_w^t}} = \frac{\int_{x_1}^{x_2} S_w dx}{x_2 - x_1}$$
(27)

here, $x_2$ and $x_1$ represent the positions of the injection well and the production well of the reservoir, respectively.

Since the distance that a given water saturation advances in the formation is proportional to $f'_w$, Equation 27 can be rewritten as:

$$\overline{S_w^{f_w^t}} = \frac{\int_{f'_w(S_{w1})}^{f'_w(S_{w2})} S_w df'_w}{f'_w(S_{w2}) - f'_w(S_{w1})}$$
(28)

By performing integration by parts:

$$\overline{S_w^{f_w^t}} = \frac{S_{w2}f'_w(S_{w2}) - S_{w1}f'_w(S_{w1}) - \int_{S_{w1}}^{S_{w2}} dS_w}{f'_w(S_{w2}) - f'_w(S_{w1})}$$
(29)

Since $S_{w1} = S_{w\max}$, $f_w(S_{w1}) = 1$ and $f'_w(S_{w1}) = 0$, therefore:

$$\overline{S_w^{f_w^t}} = \frac{S_{w2}f'_w(S_{w2}) - [f_w(S_{w2}) - 1]}{f'_w(S_{w2})}$$
(30)

For planar radial flow:

$$\frac{dr}{dt} = \frac{-Q}{\varphi A(r)}f'_w(S_w)$$
(31)

where $A(r)$ is the seepage cross-sectional area, $A(r) = 2\pi rh$.

Separating variables and integrating, we obtain:

$$-\int_{R_o}^{r} 2\pi\varphi hr dr = f'_w(S_w)\int_0^t Qdt$$
(32)

The radial flow equation for the movement of iso-saturation surfaces can be written as:

$$r_f{}^2 - r_w{}^2 = \frac{f'_w(S_w)}{\pi \varphi h} \int_0^t Q dt = \frac{f'_w(S_w)}{\pi \varphi h} W_i \tag{33}$$

where $W_i$ is the cumulative injected water volume, $W_i = \int_0^t Q dt$.

From this, we obtain:

$$f'_w(S_w) = \frac{(R^2 - r^2)\pi \varphi h}{\int_0^t Q dt} = \frac{1}{Q_i} \tag{34}$$

where $Q_i$ is the cumulative injection pore volume multiplier.

By substituting $f'_w(S_{w2})$ into Equation 30, we get:

$$\overline{S_w^{f_w}} = S_{w2} + Q_i \left[1 - f_w(S_{w2})\right] \tag{35}$$

Then, based on $\overline{S_w^{f_w}}$ and the relative permeability curves, the relative permeabilities of oil and water at the corresponding water saturation $K_{ro}(\overline{S_w^{f_w}})$ and $K_{rw}(\overline{S_w^{f_w}})$ can be obtained.

The corresponding seepage resistance for the high-permeability layer is:

$$R_H{}' = R_{wo} = \frac{R_w R_o}{R_w + R_o} = \frac{\ln \frac{r_e}{r_w}}{2\pi K_H h} \frac{1}{\frac{K_{rw}(\overline{S_w^{f_w}})}{\mu_w} + \frac{K_{ro}(\overline{S_w^{f_w}})}{\mu_o}} \tag{36}$$

after the waterflood front breaks through the production well, the further advancement distance of the waterflood front is relatively small, resulting in minimal changes in seepage resistance. Therefore, the seepage resistance in the low-permeability layer is approximately equal to the seepage resistance in the low-permeability layer at the time of breakthrough, i.e.,

$$R_L{}' = R_L^i \tag{37}$$

By separating variables and integrating the Buckley-Leverett equation, it is assumed that the waterflood front in the high-permeability layer continues to advance until the water saturation at the original production well interface reaches 98%. The corresponding advancement radius of the waterflood front in the high-permeability layer at this point is

$$r_f{}^2 - r_w{}^2 = \frac{f'_w(S_w)}{\pi \varphi h} \int_0^t Q dt = \frac{f'_w(S_w)}{\pi \varphi h} Q_i \pi (r_e{}^2 - r_w{}^2)\varphi h = f'_w(S_{wf}) Q_i (r_e{}^2 - r_w{}^2) \tag{38}$$

i.e.,

$$r_{fH}{}^2 = \frac{f'_w(S_{wf})}{\pi \varphi h} \int_0^t Q dt = \frac{f'_w(S_{wf})}{\pi \varphi h} Q_i \pi (r_e{}^2 - r_w{}^2)\varphi h = f'_w(S_{wf}) Q_i (r_e{}^2 - r_w{}^2) + r_w{}^2 \tag{39}$$

The initial seepage resistance ratio between the low-permeability and high-permeability layers is:

$$C_{RL}^{1'} = \frac{R_L{}'}{R_H{}'} = \frac{R_L^i}{R_H{}'} \tag{40}$$

The initial iterative advancement area of the waterflood front in the low-permeability layer $S_{fL}^{1'}$ is:

$$S_{fL}^{1'} = \frac{S_e}{C_{RL}^{1'}} = \pi r_{fH}^2 \frac{R_H'}{R_L^i} \tag{41}$$

The initial iterative advancement radius of the waterflood front after breakthrough $r_{fL}^{1'}$ is:

$$r_{fL}^{1'} = r_{fH} \sqrt{\frac{R_H'}{R_L^i}} \tag{42}$$

The seepage resistance in the low-permeability layer is:

$$R_L^{1'} = \frac{\ln \frac{r_{fL}^{1'}}{r_w}}{2\pi K_L h} \frac{1}{\frac{K_{rw}(\overline{S_w^{Lwo}})}{\mu_w} + \frac{K_{ro}(\overline{S_w^{Lwo}})}{\mu_o}} + \frac{\mu_o \ln \frac{r_e}{r_{fL}^{1'}}}{2\pi K_L h K_{ro}(S_{wc})} \tag{43}$$

Repeat the iterative calculations until the relative error between two consecutive iterations is less than 0.1%. The calculated advancement radius of the waterflood front in the low-permeability layer ($r_{fL}^{1'}$) will be the advancement radius of the waterflood front in the low-permeability layer when the waterflood front in the high-permeability layer breaks through at the production well.

As mentioned earlier, when there are more than two heterogeneous reservoir layers with different permeabilities, the calculation method for the post-breakthrough water drive front remains applicable.

Based on the previously calculated advancement radius of the waterflood front in the low-permeability region, the volume fraction of the weak water-flushed zone in the low-permeability layer at the corresponding time can be obtained:

$$\varphi_B = \frac{r_e^2 - r_{fL}^2}{r_e^2} \times 100 \tag{44}$$

## 2.2. Prediction and calculation method for interlayer weak water-flushed zone distribution

Based on the assumption conditions established in the previous model, using reservoir parameter data from a specific well in the Gudong experimental area, a corresponding simulation of the waterflooding process in heterogeneous water-flooded oil reservoirs was conducted using the CMG numerical model. The simulation was performed using reservoir parameter data from a well in the Gudong test area to observe the advancement of the water flooding front in the weak water-washed zone. The reservoir adopts a five-spot injection-production well pattern, consisting of four production wells and one injection well. The average distance between the injection and production wells is 150 meters. The overall equivalent permeability of the high-permeability section of the reservoir is approximately $2000 \times 10^{-3} \mu m^2$, while the overall equivalent permeability of the low-permeability section is approximately $200 \times 10^{-3} \mu m^2$. The thickness of the high and low permeability layers is about 1.5 meters. The crude oil viscosity at reservoir temperature is 65.64 mPa·s, and the viscosity of the displacement water is 0.57 mPa·s. The relative permeability data is shown in Fig 2, and the numerical simulation model for water flooding is illustrated in Fig 3.

Starting from the initial reservoir conditions, water flooding is conducted, and the production data from the water flooding development stage in the Gudong test area are used for history matching of the water cut and oil production rate in the block. The history matching results are shown in Fig 4. From the data trends in the figure, it can be observed that due to the stratified treatment of permeability in high- and low-permeability reservoir layers, the numerical simulation results exhibit a relatively stable pattern compared to the field production data. However, the overall correlation remains high, and the data trends are consistent. This ensures that the simulation model meets the requirements for subsequent displacement effect comparison and evaluation.

To further verify the applicability of the calculation method in cases where the number of heterogeneous reservoir layers with different permeabilities exceeds two, a numerical simulation model was established based on the same reservoir parameter conditions. This model includes three heterogeneous reservoir layers with different permeability values. In this model, the

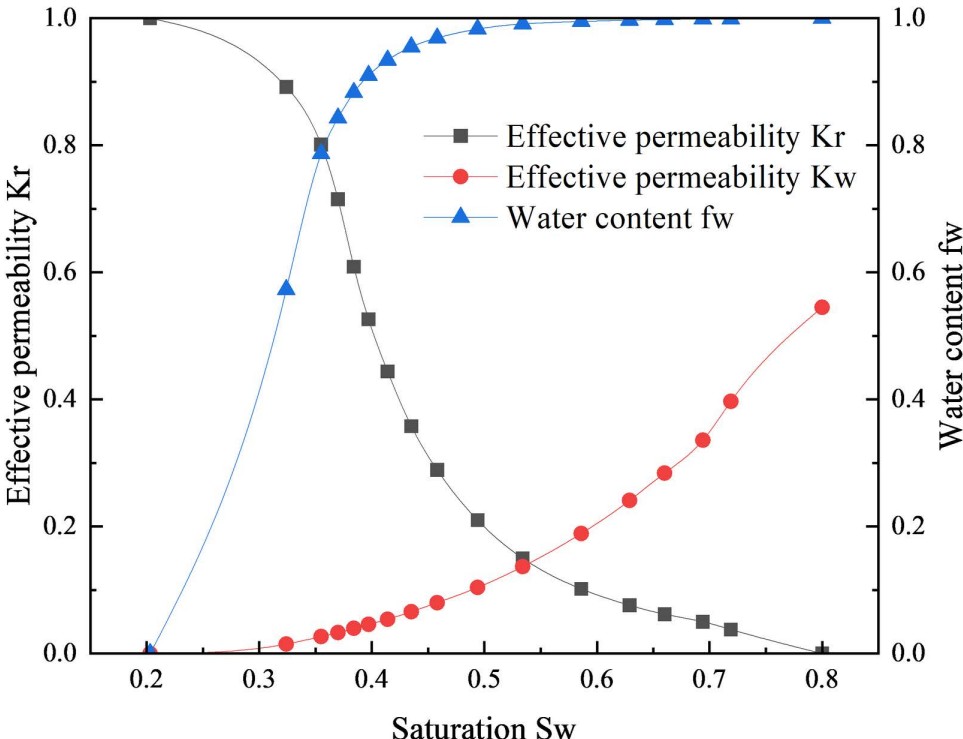

**Fig 2. Relative permeability curve of a well in Gudong test area.**

permeability values of the high-permeability layer, medium-high-permeability layer, and low-permeability layer are 2000, 1000, and 200×10⁻³ μm², respectively. The schematic diagram of the three-layer numerical simulation model is shown in Fig 5.

## 3. Experimental results and discussion

Based on the model assumptions described earlier and using the reservoir parameter data from a well in the Gudong test area, the CMG numerical simulation software was used to simulate the water flooding process in a heterogeneous reservoir. The focus was on observing the advancement of the water flooding front in the weak water-washed zone. The iterative calculation method described above was employed, and the results were compared with those obtained from physical simulation experiments and numerical simulation experiments. This comparison was conducted to validate the accuracy and applicability of the calculation method.

### 3.1 Prediction model calculation results

(1)     When the Waterflood Front Breaks Through the Production Well

According to the relative permeability curve data mentioned above, the saturation of the waterflood front, $S_{wf} = 0.35$. For a vertical double-layer heterogeneous model, the injected water preferentially breaks through the high-permeability layer. When the injected water breaks through the production well of the high-permeability layer, the average water saturation of the high-permeability layer ($\overline{S_W^H}$) is obtained from the relative permeability curve ($\overline{S_W^H} = 0.4$). At this point, the flow resistance of the high-permeability layer is

$$R_H = \frac{R_w R_o}{R_w + R_o} = \frac{\ln \frac{r_e}{r_w}}{2\pi K_H h} \frac{1}{\frac{K_{rw}(\overline{S_w^H})}{\mu_w} + \frac{K_{ro}(\overline{S_w^H})}{\mu_o}} = 2.19 \times 10^{-2}(mPa \cdot s)/(\mu m^2 \cdot cm)$$

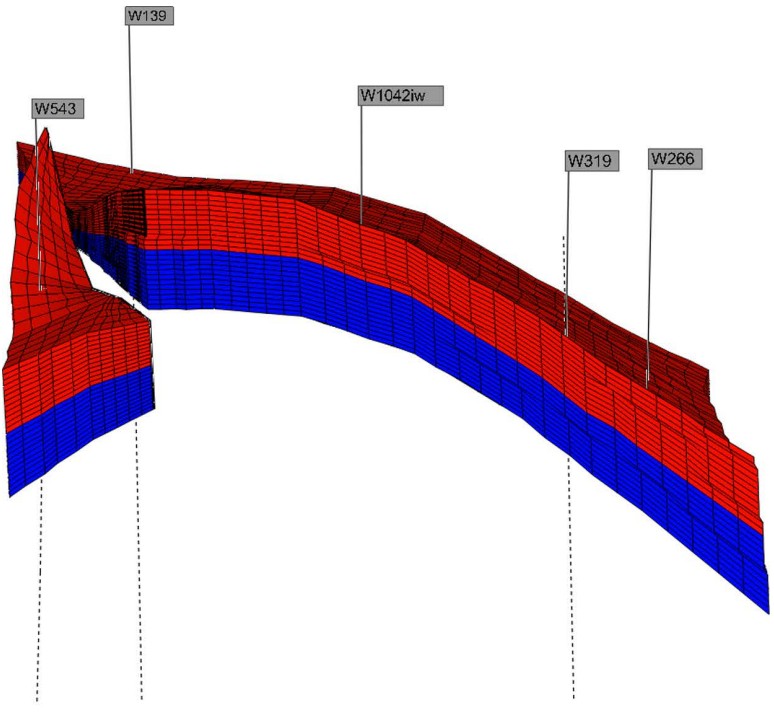

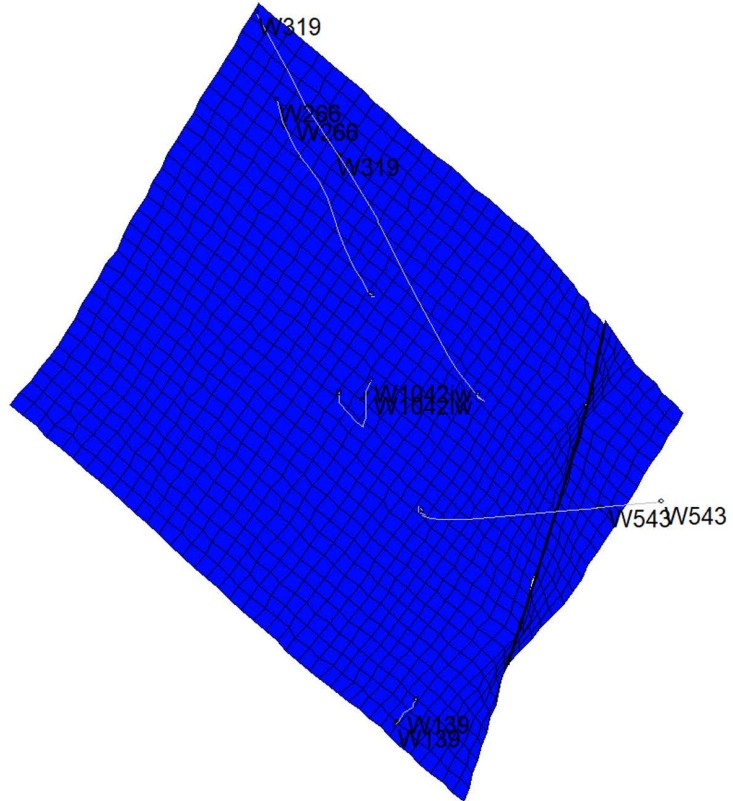

**Fig 3. Schematic diagram of numerical simulation model.**

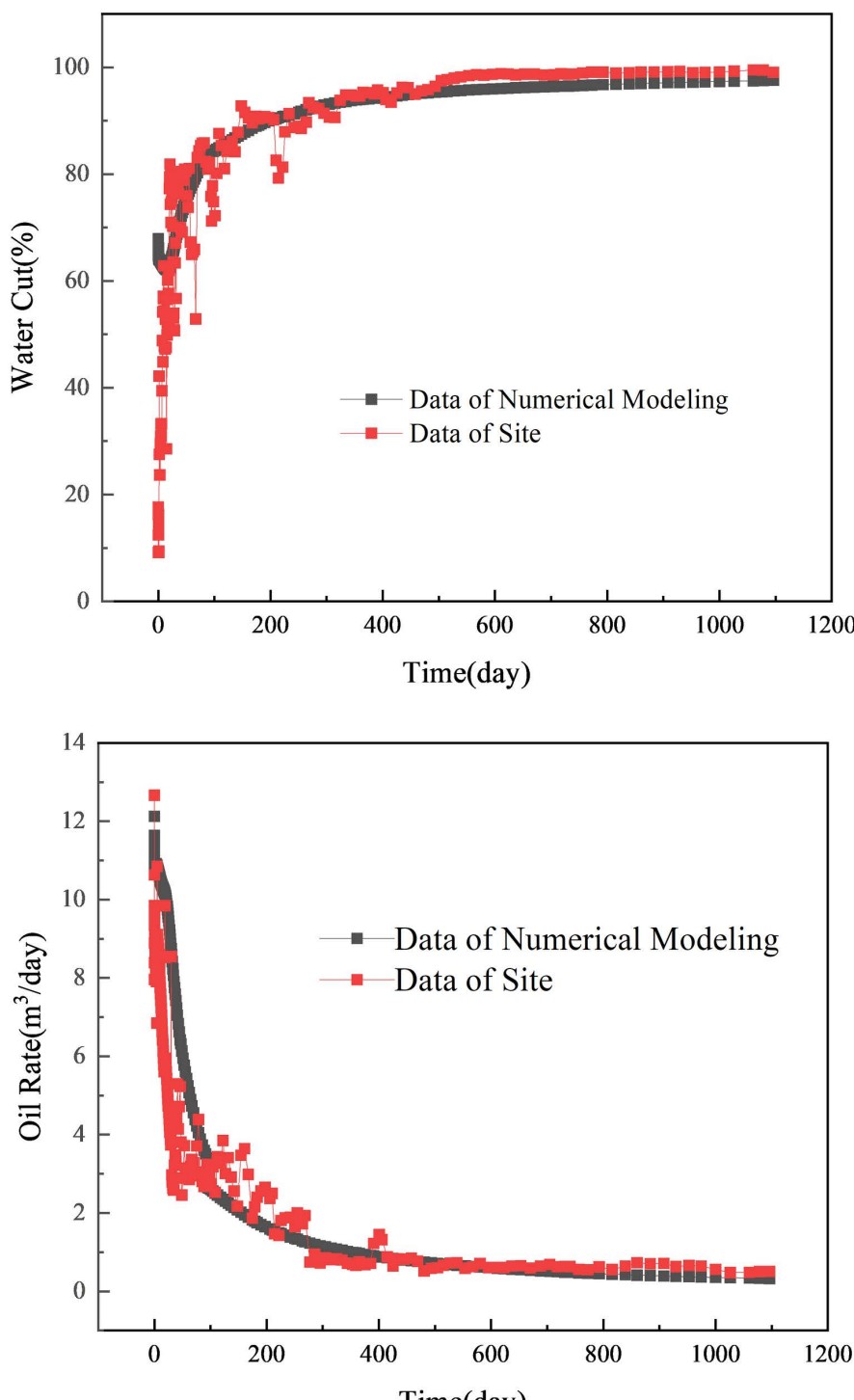

**Fig 4. History matching of the water cut and oil production rate.**

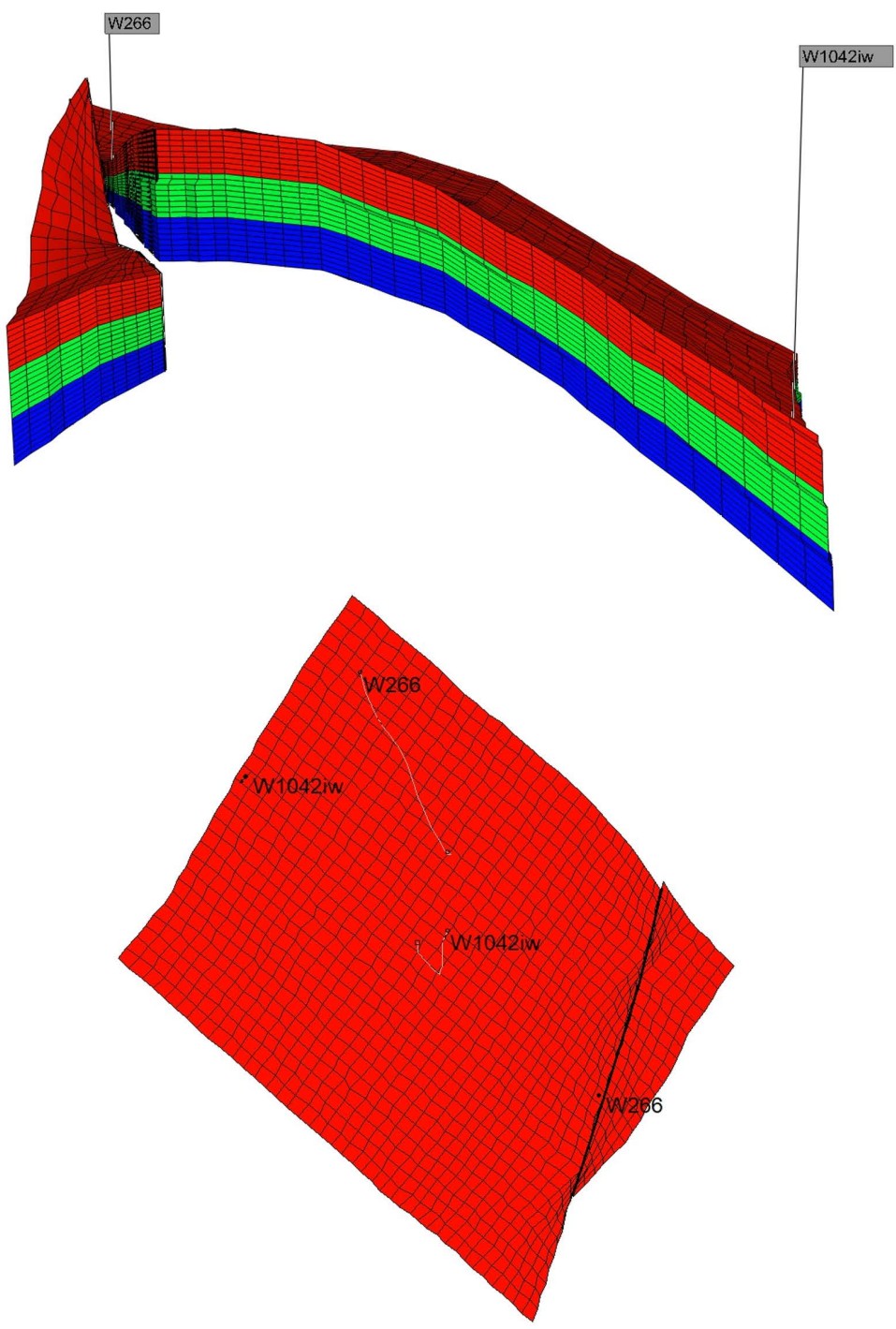

**Fig 5. Schematic diagram of three-layer numerical simulation model.**

From the start of waterflooding to breakthrough, the advancement radius of the waterflood front in the high-permeability layer ($r_{fH}$) is 150m.

The waterflooding process in the model is characterized by a constant displacement rate. Since the flow velocity in the high-permeability layer is greater than that in the low-permeability region, the ratio of the average water saturation in the high-permeability layer to the average water saturation in the low-permeability layer ($\frac{\overline{S_w^H}}{\overline{S_w^L}} \geq 1$) increases as displacement proceeds before the waterflood front breaks through the oil well. Consequently, the flow resistance ratio between the low- and high-permeability layers ($\frac{R_L}{R_H} \geq 10$) also increases.

Maximum advancement radius in the low-permeability region:

when $\frac{R_L}{R_H} = 10$, $r_{fL}^{max} = \sqrt{\frac{K_L}{K_H}} \cdot r_e = 47.43m$.

Using the iterative method described earlier, we calculated the advancing distance of the waterflooding front in the low-permeability layer, the results shows that when the waterflooding front breaks through at the production well in the high-permeability zone, the advancing radius of the waterflooding front in the low-permeability zone is 30.82 m, with a distance of 119.18cm from the production well. The volume proportion of the weak water-flushed zone in the low-permeability layer is 95.78%.

(2) After the Waterflood Front Breaks Through the Production Well

The water cut $f_w$ is calculated based on Equation (23) and relative permeability curve data. The data results are fitted using Equation (26) to obtain parameters a = 6.657 × 103 and b = 15.744, that is:

$$\ln\left(\frac{1-f_w}{f_w}\right) = \ln\left(\frac{6.657 \times 10^3 \mu_w}{\mu_o}\right) - 15.744 S_w$$

When the water cut at the production well $f_{w2} = 98$ %, the water saturation at the production well $S_{w2}$ can be calculated as 0.505 according to the above formula. Based on Equation (35), when the water cut at the production well is 98%, the average water saturation in the high-permeability layer is:

$$\overline{S_w^{fl}} = S_{w2} + Q_i\left[1 - f_w(S_{w2})\right] = 0.505 + 1.86 \times (1 - 0.98) = 0.542$$

Therefore, according to Equation (36), the flow resistance of the high-permeability layer at this water cut is:

$$R_H = \frac{R_w R_o}{R_w + R_o} = \frac{\ln\frac{r_e}{r_w}}{2\pi K_H h} \frac{1}{\frac{K_{rw}(\overline{S_w^{fl}})}{\mu_w} + \frac{K_{ro}(\overline{S_w^{fl}})}{\mu_o}} = 1.04 \times 10^{-2}(mPa \cdot s)/(\mu m^2 \cdot cm)$$

Using the iterative method described earlier, the calculation of the advancing distance of the waterflooding front in the low-permeability layer shows that when the production well has a water cut of 98%, the advancing distance of the waterflooding front in the low-permeability zone is 77.35 m, with a distance of 88.46 m from the production well. The volume proportion of the weak water-flushed zone in the low-permeability layer is 73.41%.

## 3.2. Numerical simulation verification

A water flooding simulation was conducted using the numerical model established in Section 2.2. When the water cut at the outlet end of the high-permeability layer reached the critical water saturation of the water flooding front (i.e., water saturation exceeding 35%), the water flooding front broke through in the high-permeability layer at the oil well. At this stage, the distribution of remaining oil in the high- and low-permeability layers is shown in Fig 6.

After the water flooding front breaks through at the oil well, the water cut at the production end of the reservoir continues to increase. The simulation was further extended until the cumulative injected pore volume (PV) reached 1.5, at which

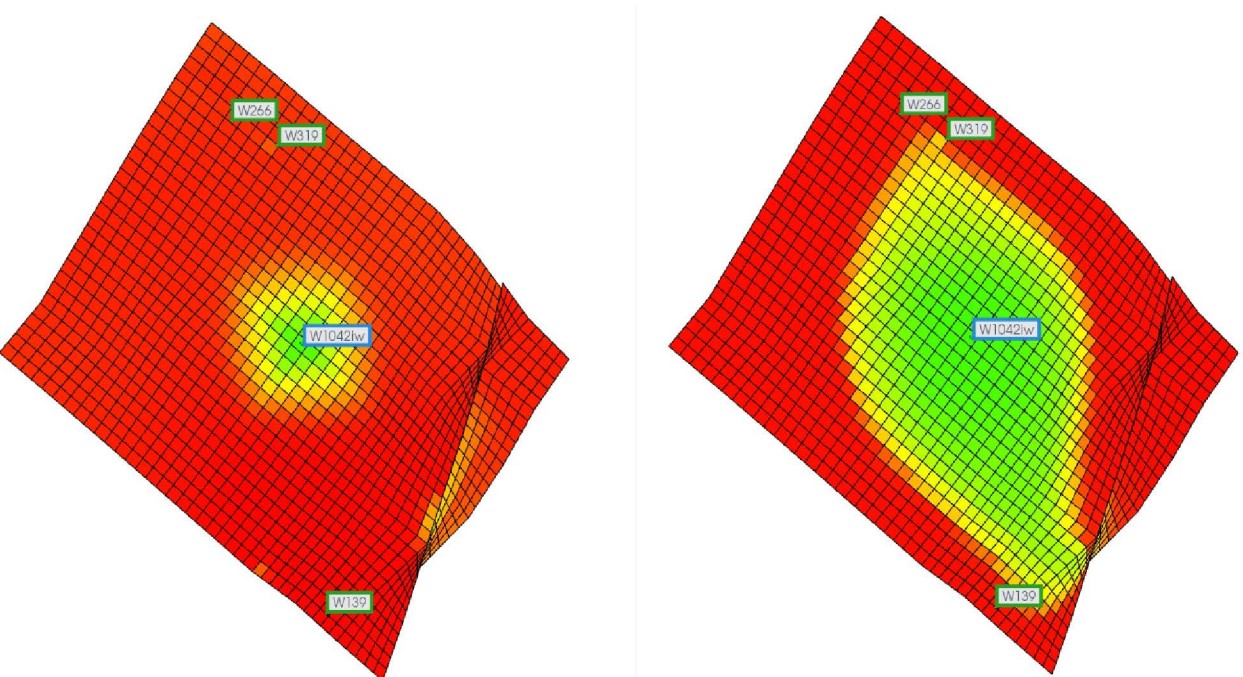

**Fig 6. The distribution of residual oil in the numerical simulation model when the water drive front breaks through the well.** (a) low-permeability layer (b) high-permeability layer.

point the water cut of the reservoir exceeded 98%. The remaining oil distribution in the high- and low-permeability layers at this stage is shown in Fig 7.

According to the prediction calculation method in this paper, when the waterflood front breaks through from the high-permeability layer to the production well, the advancing distance of the waterflood front in the low-permeability layer is 30.82 m. A comparison is made with the displacement results from the CMG numerical simulation model, based on the volume ratio of the weak water-flushed zone at the waterflood front saturation and water saturation. When the waterflood front in the high-permeability layer reaches the production well in the numerical simulation model, the average advancing radius of the waterflood front in the low-permeability layer is about 35m. At this point, the volume ratio of the weak water-flushed zone in the low-permeability layer is 94.55%, with a relative error of 1.22% compared to the calculated result. When the water cut at the production well of the high-permeability layer reaches 98%, the advancing distance of the waterflood front in the low-permeability layer is 77.35m. A comparison with the CMG numerical simulation model shows that when the water cut at the production well of the high-permeability layer reaches 98%, the average advancing radius of the waterflood front in the low-permeability layer is about 70 m. At this point, the volume ratio of the weak water-flushed zone in the low-permeability layer is 78.22%, with a relative error of 4.81%. During the calculation using the new predictive model, factors such as capillary pressure, gravity segregation, and the compressibility of rock and fluid were neglected, which led to some deviations from the numerical simulation results. However, as shown by the computed results, the relative error in all cases remained below 5%, which meets the required accuracy for field applications. Thus, based on the comparison between CMG simulation results and the new predictive model, we conclude that the proposed weak water-flushed zone distribution prediction model exhibits sufficient accuracy and practical applicability for field use.

To further verify the applicability of the weak water-flushed zone distribution prediction model under different geological conditions, numerical simulation models were used to vary the permeability ratio and the oil-water viscosity ratio. The

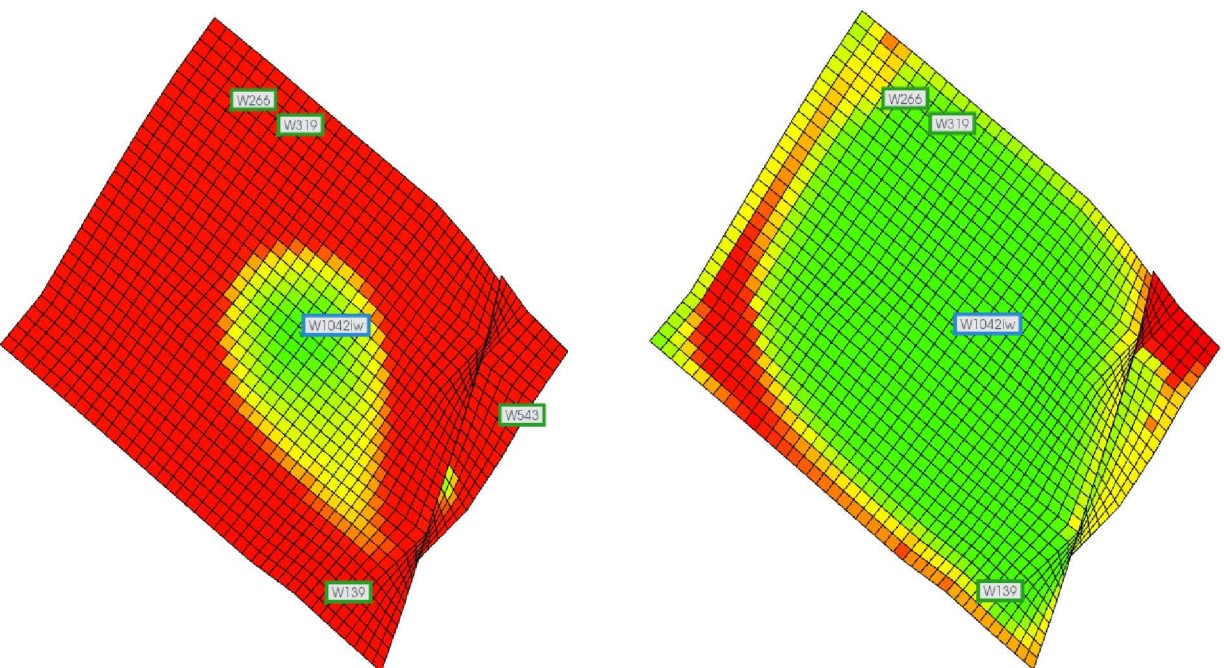

**Fig 7. The distribution of residual oil in the numerical simulation model when the water cut reaches 98%.** (a) low-permeability layer (b) high-permeability layer.

iterative calculation method of the predictive model was then applied to determine the volume proportion of the weak water-flushed zone in the low-permeability layer. A comparison of the computed results with the numerical simulation results is presented in Table 1.

From the data in Table 1, it can be observed that after varying the reservoir permeability ratio and oil-water viscosity ratio, the discrepancy between the calculated and simulated values for the volume proportion of the weak water-flushed zone in the low-permeability reservoir still falls within the acceptable practical accuracy range. This confirms that the distribution prediction model is applicable for reservoir calculations under different geological conditions.

For heterogeneous reservoirs with more than two layers exhibiting different permeability characteristics, the distribution prediction model is applied to high-permeability and medium-permeability reservoirs, as well as high-permeability and low-permeability reservoirs, based on the reservoir and displacement parameters of a three-layer numerical simulation model. Thus we got the volume fraction of weak water-flushed zones within the medium- and low-permeability reservoirs.

The numerical simulation results were obtained using a three-layer numerical simulation model and compared with the calculated results. The distribution of weak water-flushed zones obtained from the numerical simulation is shown in Fig 8.

According to the calculation results, when the water cut reaches 98%, the volume proportion of the weak water-flushed zone in the medium-permeability reservoir is 49.28%, while in the low-permeability reservoir, it is 73.41%. Referring to Fig 8,

**Table 1. Comparison of experimental results under different geological conditions.**

| Permeability | Oil viscosity / mPa·s | Calculation results/% | Simulation results/% | Relative error/% |
|---|---|---|---|---|
| 2000/200 | 65.64 | 73.41 | 78.22 | 4.81 |
| 1000/200 | 65.64 | 49.28 | 51 | 1.72 |
| 2000/200 | 150 | 76.02 | 79.45 | 3.43 |
| 1000/200 | 150 | 54.28 | 55.56 | 1.28 |

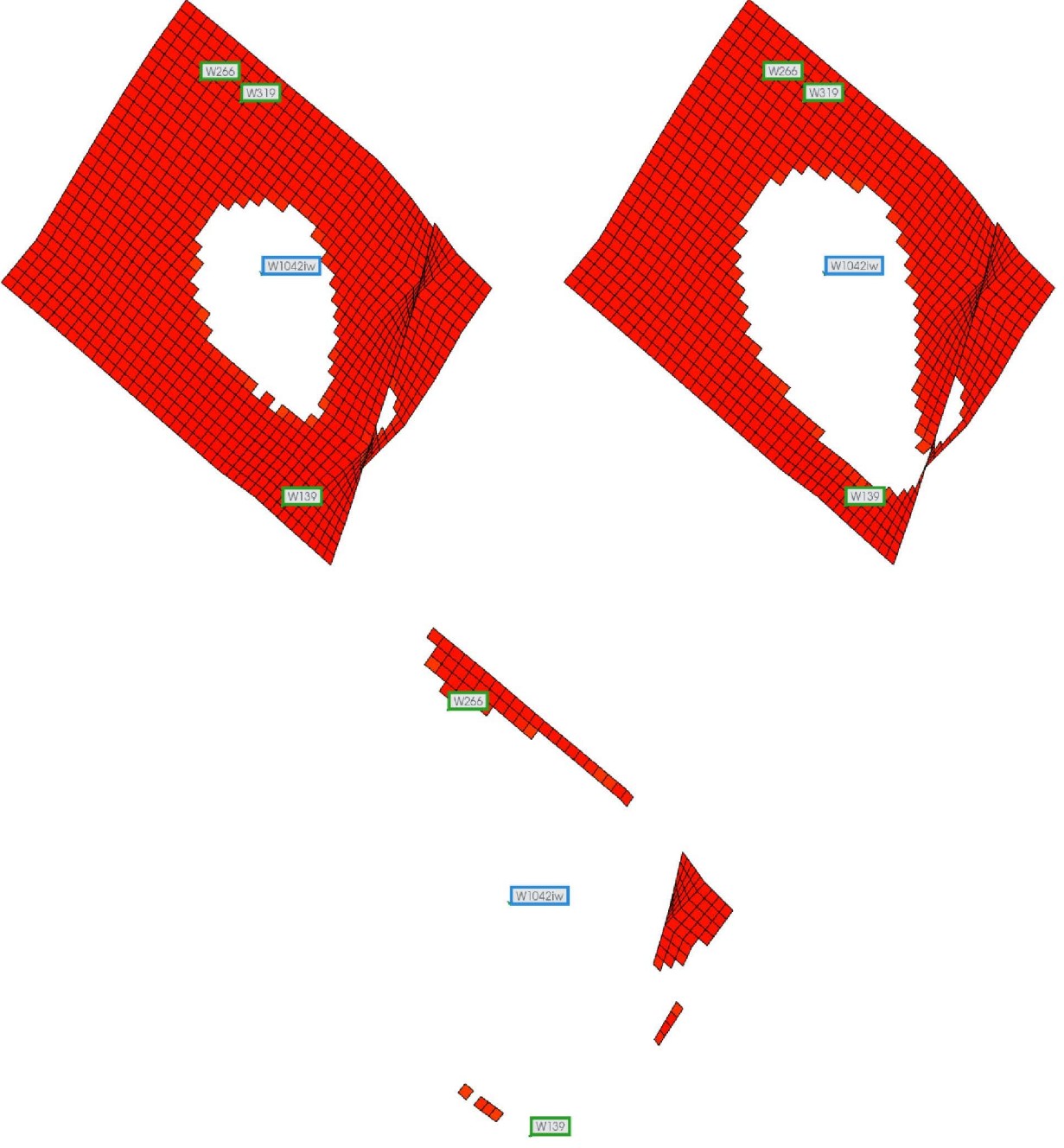

**Fig 8. The distribution of weak water-flushed zones of three-layer numerical simulation model.** (a) low-permeability layer (b) mid-permeability layer (c) high-permeability layer.

which illustrates the distribution of the weak water-flushed zone in the post-flooding numerical simulation model, the corresponding values for the medium-permeability and low-permeability reservoirs are 51% and 78.22%, respectively. The relative errors between the calculated and simulated results are 1.72% and 4.81%, respectively. These results indicate that the distribution prediction model remains applicable even when the number of heterogeneous reservoir layers exceeds two.

The experimental results confirm that the calculation method for predicting the distribution of the weak water-flushed zone in heterogeneous water-flooded reservoirs—derived from Darcy's Law and Buckley-Leverett's non-piston-like water flooding theory—demonstrates high accuracy and applicability under different geological conditions. The predicted volume proportions of the weak water-flushed zone in various layers of multi-layer heterogeneous reservoirs show only minor relative errors compared to actual water flooding results, further validating the practical reliability of the model.

## 4. Applications example of the calculation model

Using the weak water-flushed zone distribution prediction method described in this paper, and based on the data from a well in the Gudong experimental area as previously mentioned, the actual reservoir extraction process can be calculated and predicted. This allows for the estimation of the volume proportion of the weak water-flushed zone in the low-permeability layer at different water cut stages. Additionally, the model can be used to study and calculate the limiting and initiating conditions for the formation of the weak water-flushed zone. Based on these findings, corresponding reservoir management strategies can be proposed to further enhance the oil recovery rate.

### 4.1. Calculating the volume of the weak waterflooding zone in the low-permeability layer at different water cut stages

According to Equation 39, the advancing distance of the high-permeability layer is calculated for each water cut stage. When the advancing radius of the high-permeability layer exceeds 150 m, the waterflooding front reaches the production well. From the moment the waterflooding front reaches the production well until the liquid water cut at the production well reaches 98%, the volume proportion of the weak water-flushed zone in the low-permeability layer can be calculated using the prediction method described in this article.

According to the calculation results, when the water cut exceeds 78.3%, the waterflooding front in the high-permeability layer advances beyond a radius of 150 meters, reaching the production end. At this point, the weak water-flushed zone in the low-permeability layer accounts for 93.96% of the total volume. Similarly, the volume fraction of the weak water-flushed zone at different water cut stages is computed after the waterflooding front in the high-permeability layer reaches the production end. The calculation results are presented in Table 2 and Fig 9.

From the Table 2 and Fig 9, it can be observed that under the conditions of this reservoir, when the water cut reaches approximately 78.3%, the waterflood front in the high-permeability layer reaches the production well of the reservoir. At this point, the volume ratio of the weak water-flushed zone in the low-permeability layer is 93.96%, and it gradually decreases as the water cut increases. When the reservoir enters the high water cut stage with a water cut greater than 90%, the volume ratio of the weak water-flushed zone in the low-permeability layer shows a noticeable decrease. When the water cut reaches 98%, the volume ratio of the weak water-flushed zone decreases to 73.41%.

### 4.2. Limit conditions for the formation of the weak waterflooding zone

In the Gudong experimental area, the oil viscosity at the reservoir temperature is 65.64 mPa·s, and the viscosity of the displacing water is 0.57 mPa·s. Under these oil-water viscosity ratio conditions, the limit conditions for the formation of the weak water-flushed zone in the low-permeability strip of the reservoir can be calculated using the method described in this paper. We use the new predictive calculation model to calculate the volume fraction of weak water-flushed zones at different permeability ratio. When the permeability ratio gradually decreases to a certain value, weak water-flushed zones no longer exist in low-permeability layers. The calculation results are presented in Table 3 and Fig 10.

According to the data presented in Table 3 and Fig 10, when the permeability ratio is 10:1, the weak water-flushed zone in the low-permeability layer accounts for 95.78% of the volume at the moment when the waterflooding front in the high-permeability layer breaks through to the production end. As waterflooding continues and the water cut reaches 98%, the volume fraction of the weak water-flushed zone in the low-permeability layer decreases to 73.41%. Under the given

**Table 2. Results of iterative calculation of reservoir data in different water cut stages.**

| $f_w$/% | The proportion of weak water-flushed zone/% |
|---|---|
| 78.3 | 93.96 |
| 80 | 93.87 |
| 85 | 93.44 |
| 90 | 92.57 |
| 95 | 89.69 |
| 98 | 73.41 |

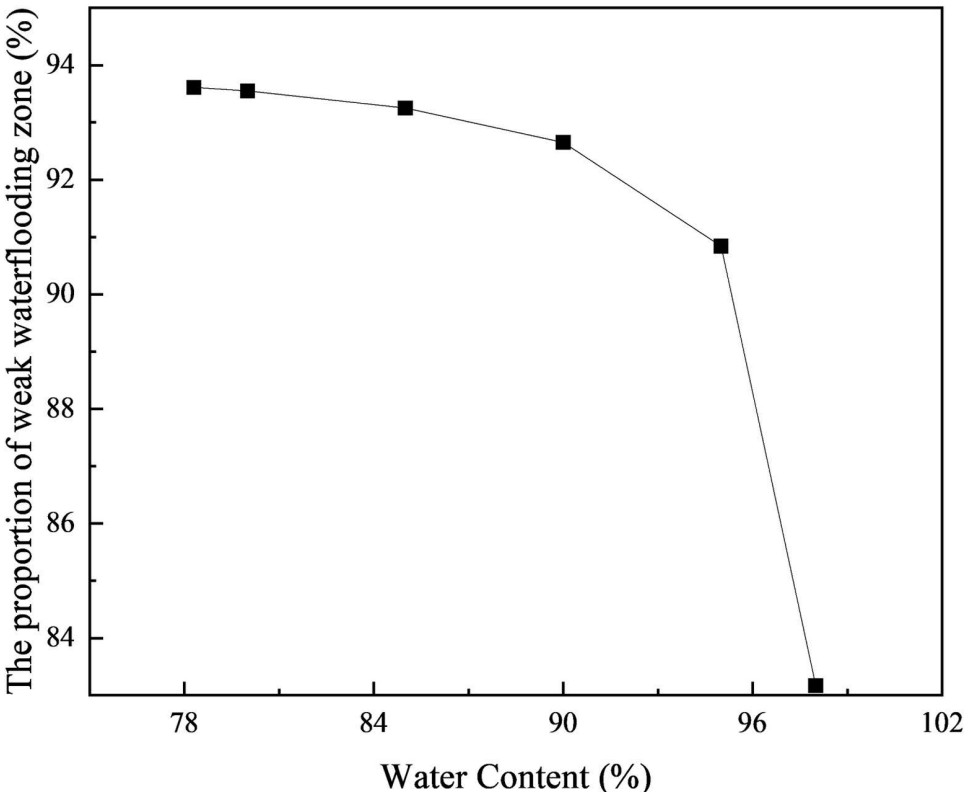

**Fig 9. Volume proportion of weak water-flushed zone in low permeability formation at different water cut stages.**

oil-water viscosity ratio, when the permeability ratio is large, the volume fraction of the weak water-flushed zone gradually decreases with a reduction in permeability ratio after the waterflooding front in the high-permeability layer breaks through. When the permeability contrast is small, the decline in the weak water-flushed zone volume fraction becomes more pronounced as the contrast decreases. However, when the water cut exceeds 98%, the weak water-flushed zone volume fraction decreases more rapidly and significantly with the reduction in permeability ratio. When the permeability ratio decreases to 2.4:1, iterative calculations fail to converge, indicating that at a water cut of 98%, the waterflooding front also breaks through from the production end in the low-permeability layer, and the weak water-flushed zone no longer exists.

Based on the above calculation results, under the reservoir conditions of the Gudong test area, when the permeability ratio between high- and low-permeability layers is less than 2.4:1 and the water cut reaches 98%, the weak water-flushed

 

**Table 3. Volume proportion of weak water-flushed zone in low permeability formation at different permeability ratio.**

| High-permeability layer permeability /10⁻³μm² | Permeabil-ity ratio | Advance radius of water drive front when the high-permeability layer breakthrough/cm | The proportion of weak water-flushed zone/% | Advance radius of water drive front at water content 98%/cm | The proportion of weak water-flushed Zone/% |
|---|---|---|---|---|---|
| 2000 | 10 | 30.82 | 95.78 | 77.35 | 73.41 |
| 1500 | 7.5 | 36.94 | 93.93 | 88.44 | 65.24 |
| 1000 | 5 | 47.95 | 89.77 | 106.83 | 49.28 |
| 800 | 4 | 55.53 | 86.28 | 118.56 | 37.53 |
| 500 | 2.5 | 76.33 | 73.79 | 147.7 | 36.51 |
| 480 | 2.4 | 78.52 | 72.60 | 150 | 0 |
| 200 | 1 | 150 | 0 | 150 | 0 |

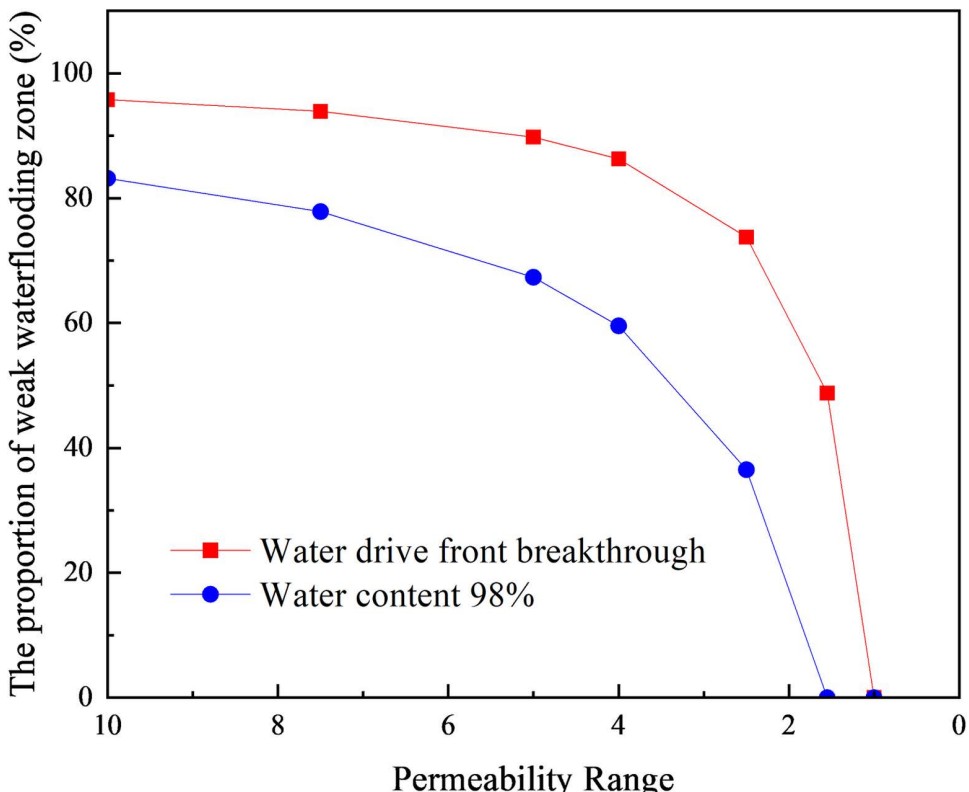

**Fig 10. The volume proportion of weak water-flushed zone in low permeability layer varies with the permeability ratio.**

zone in the low-permeability layer ceases to exist. Therefore, the critical condition for weak water-flushed zone formation is that the permeability ratio between high- and low-permeability layers must be greater than 2.4:1.

### 4.3. Study on the conditions for initiating the weak waterflooding zone

In the actual oilfield production process, after primary and secondary oil recovery phases, when the water cut exceeds 98%, the reservoir enters a stage of very high water cut. To address this, measures such as plugging the high-permeability layers or increasing the viscosity of the displacing fluid, or altering the oil-water viscosity ratio within the reservoir, can be taken to mobilize the remaining oil in the weak water-flushed zone. In this scenario, the impact of either

plugging the high-permeability layers or increasing the viscosity of the displacing fluid on the weak water-flushed zone can be calculated and predicted separately. Based on these predictions, the more effective and economically beneficial tertiary oil recovery enhancement measures can be selected to improve production.

**4.3.1. Impact of high-permeability layer plugging on the initiation of the weak waterflooding zone.** Based on the previously described reservoir parameters and waterflooding results, when the water cut reaches 98% during the secondary oil recovery waterflooding process, with the waterflood front in the low-permeability layer advancing to a position of 77.35 m, plugging is performed on the reservoir to adjust the profile. Waterflooding continues in the subsequent extraction process until the water cut reaches 98% again. Under these conditions, the area proportion of the weak water-flushed zone is calculated to observe the impact of the degree of plugging in the high-permeability layer on the initiation of the weak water-flushed zone. The calculation results are presented in Table 4 and Fig 11.

According to the calculation results, it can be seen that the volume ratio of the weak water-flushed zone in the low-permeability layer decreases as the degree of blockage in the high-permeability layer increases, as shown in Table 4 and Fig 11. When the degree of blockage in the high-permeability layer is relatively low, the increase in the initiation ratio of the weak water-flushed zone in the low-permeability layer is relatively slow as the degree of blockage increases. However, when the degree of blockage in the high-permeability layer is relatively high, the initiation ratio of the weak water-flushed zone in the low-permeability layer increases rapidly as the degree of blockage increases. When the permeability of the high-permeability layer decreases to $480 \times 10^{-3} \mu m^2$, the initiation ratio of the weak water-flushed zone reaches 100%, the volume decreases to zero, and the waterflood front in the low-permeability layer breaks through from the production well, with no weak water-flushed zone remaining in the reservoir.

**4.3.2. Impact of high-permeability layer plugging on the initiation of the weak water-flushed zone.** Based on the previously described reservoir parameters and waterflooding results, when the water cut reaches 98% during the secondary oil recovery waterflooding process, with the waterflood front in the low-permeability layer advancing to a position of 77.35 m, a polymer is added to the displacing water to alter its viscosity. The subsequent extraction continues until the water cut reaches 98% again. Under these conditions, the area proportion of the weak water-flushed zone is calculated, and the impact of increasing the displacing fluid viscosity on the initiation of the weak water-flushed zone is observed. The calculation results are shown in Table 5 and Fig 12.

Based on the calculation results, it can be observed that the proportion of the weak water-drive zone in low-permeability layers increases rapidly with the viscosity of the displacement medium, as shown in Table 5 and Fig 12. When the viscosity of the displacement medium starts to increase from 0.57 mPa·s, the volume fraction of the weak water-drive zone decreases rapidly in the early stage of viscosity increase. This indicates that even a slight increase in the viscosity of the displacement medium can significantly improve the displacement efficiency in the reservoir. When the viscosity of the displacement medium reaches 10 mPa·s, the volume of the weak water-drive zone becomes zero, and the water-drive front in the low-permeability layer breaks through at the production well, meaning there is no longer any weak water-drive zone within the reservoir.

Through the above analysis of activation impacts, it is evident that by utilizing the new predictive model for weak water-flushed zone distribution, we can compare different enhanced oil recovery (EOR) measures under the same economic

**Table 4. Effect of high permeability layer plugging on volume proportion of weak water-flushed zone.**

| High-permeability layer permeability after plugging /10⁻³μm² | Advance radius of water drive front in low-permeability layer after plugging/cm | The proportion of weak water-flooding zone/% | The startup ratio of weak water-flooding zone/% |
|---|---|---|---|
| 1800 | 81.24 | 70.67 | 3.73 |
| 1500 | 88.44 | 65.24 | 11.13 |
| 1000 | 106.83 | 49.28 | 32.87 |
| 800 | 118.56 | 37.53 | 48.88 |
| 480 | >150 | 0 | 100 |

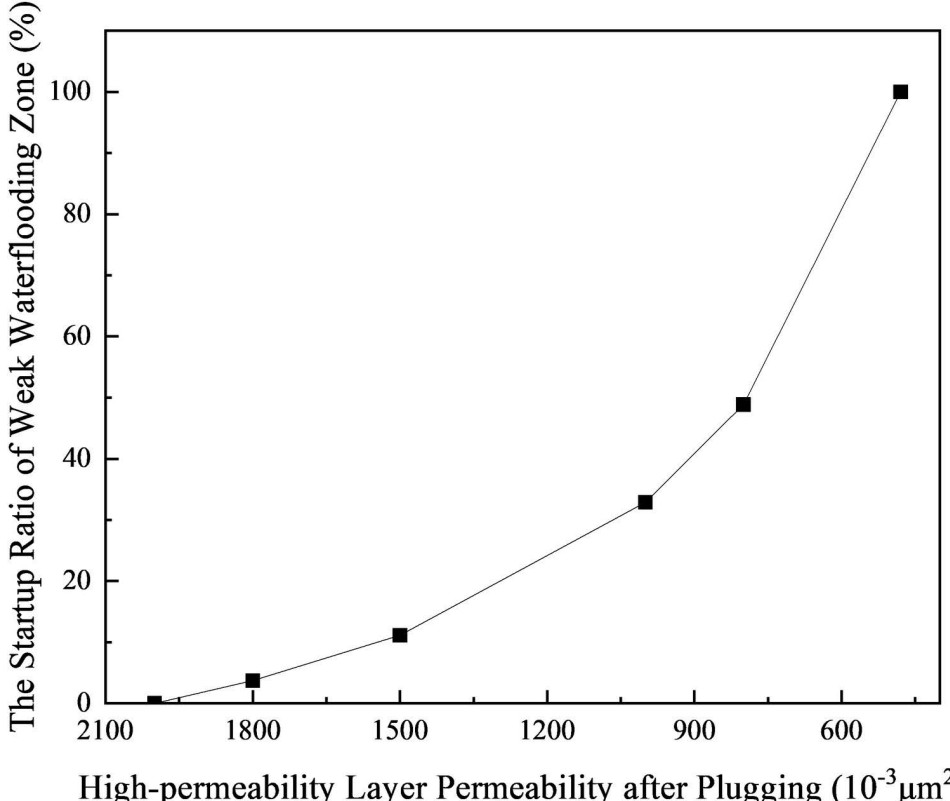

**Fig 11. The startup ratio of weak water-flushed zone in low permeability layer varies with the sealing degree of high permeability layer.**

investment to assess their effectiveness in reducing the volume proportion of the weak water-flushed zone and enhancing oil recovery. For instance, under the current production cost conditions, assuming that with the same economic investment, the high-permeability layer permeability can be reduced to 1500 md through permeability blocking, or polymer injection can be applied to increase the displacing fluid viscosity to 5 mPa·s, a comparison based on Tables 4 and 5 shows that the weak water-flushed zone activation ratios for these two methods are 11.13% and 61.94%, respectively. This indicates that, given the current reservoir and cost conditions, opting for polymer injection for enhanced oil recovery is more cost-effective than permeability blocking in the high-permeability layer, yielding a lower input-output ratio.

## 5. Findings and conclusions

(1) Existing studies heavily rely on complex physical models, involve cumbersome computational processes, and lack sufficient research on weak water-flushed zone distribution under radial flow conditions. To address these issues, this study derives waterflooding front advancement equations for different permeability zones in a heterogeneous reservoir (with interlayers) under radial flow conditions based on Darcy's law and the Buckley-Leverett non-piston-like waterflooding theory, using the flow resistance coefficient as an indicator. An iterative method has been established for calculating the advancement distance of the waterflood front in low-permeability layers under radial flow conditions, thus providing a relatively simple and convenient way to describe the distribution of residual oil in heterogeneous reservoirs;

(2) A numerical simulation model was established based on the reservoir conditions of the Gudong-7 test area, and waterflooding displacement simulations were conducted. The simulation results were history-matched with actual production data from

**Table 5. Effect of displacing medium viscosity on volume proportion of weak water-flushed zone.**

| Viscosity of displacement medium / mPa·s | Advance radius of water drive front in low-permeability layer after polymer injection/cm | The proportion of weak waterflooding zone/% | The startup ratio of weak waterflooding zone/% |
|---|---|---|---|
| 1 | 93.86 | 60.85 | 17.11 |
| 3 | 113.41 | 42.84 | 41.64 |
| 5 | 127.33 | 27.94 | 61.94 |
| 7 | 137.98 | 15.38 | 79.05 |
| 9 | 146.49 | 4.63 | 93.69 |
| 10 | >150 | 0 | 100 |

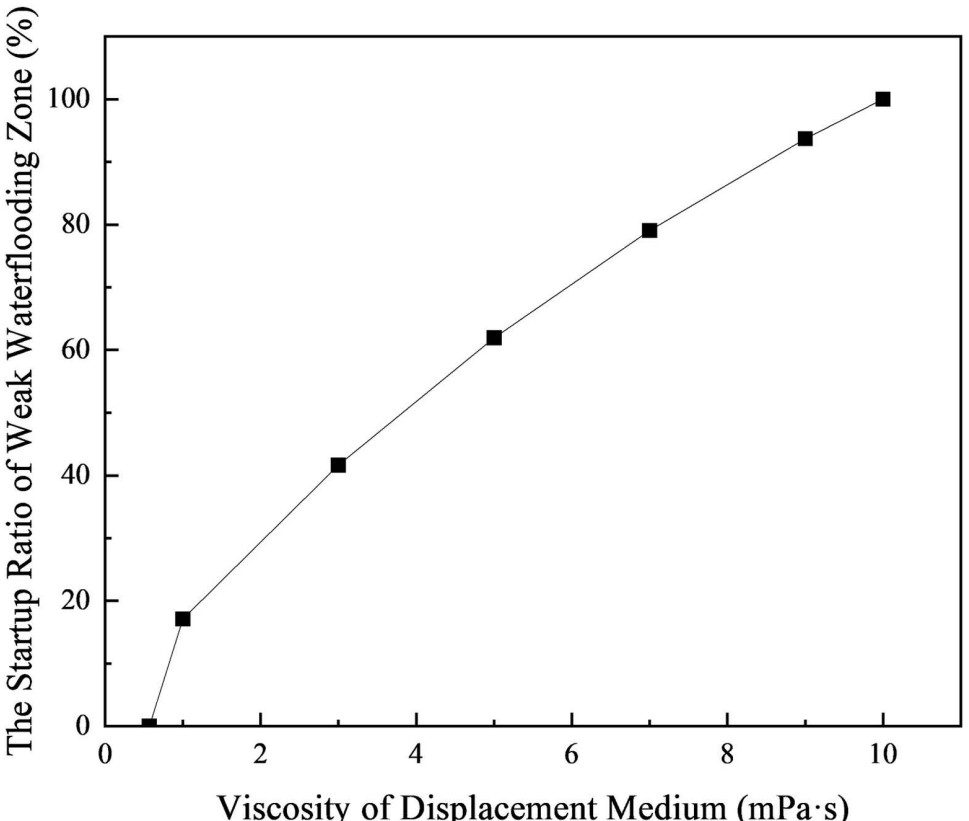

**Fig 12. The startup ratio of weak water-flushed zone in low permeability layer varies with the viscosity of displacing medium.**

the test area's waterflooding process, ensuring the applicability and accuracy of the numerical simulation experiments. The advancement of the waterflooding front in multi-layer heterogeneous reservoirs was observed through the numerical simulation tests. By comparing the simulation results with the predictions from the newly developed weak water-flushed zone distribution model, the accuracy of the proposed calculation and prediction method was validated, with the prediction error being less than 5%. This indicates that the method achieves high accuracy while maintaining computational simplicity.

(3) The established weak water-flushed zone distribution prediction method was applied to calculate the volume fraction of weak water-flushed zones in low-permeability layers at different water-cut stages. Using the weak water-flushed zone distribution prediction method established in this paper, the volume proportion of the weak water-flushed zone

in the low-permeability layer was calculated for different water cut stages. Through calculations of the distribution of weak water-flushed zones in heterogeneous reservoirs with varying permeability ratio, it was found that the limiting condition for the existence of the weak water-flushed zone in a certain well in the Goudong experimental area is that the permeability ratio of the reservoir must be greater than 2.4. Additionally, based on the previous waterflooding model, the technical limits for the startup of the weak water-flushed zone after performing profile modification and water blocking or polymer injection during the ultra-high water cut stage were calculated. When the permeability of the high-permeability layer drops to $480 \times 10^{-3} \mu m^2$ or the viscosity of the displacing medium reaches 10 mPa·s, the weak water-flushed zone in the low-permeability layer can be fully activated.

(4) After studying the interlayer distribution of weak water-flushed zones (including barriers) in heterogeneous reservoirs, further research can be conducted on the distribution of weak water-flushed zones in non-homogeneous reservoirs without barriers, exploring the influence of gravity on the remaining oil distribution when there are no barriers between different permeability layers. And study the specific distribution of weak water-flushed zones within the layers to obtain more specific and accurate predictions of remaining oil distribution.

## Supporting information

**S1 File. Symbol annotation.** Some of the symbols involved in equation derivation may not be easy to understand, so they are explained in supporting file 1.
(DOCX)

## Author contributions

**Conceptualization:** Guicai Zhang.

**Data curation:** Mengyun Li.

**Formal analysis:** Haihua Pei.

**Investigation:** Liu Yang.

**Methodology:** Mengyun Li, Guicai Zhang.

**Software:** Mengyun Li.

**Supervision:** Ping Jiang.

**Validation:** Ping Jiang, Liu Yang.

**Writing – original draft:** Mengyun Li.

**Writing – review & editing:** Haihua Pei.

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
