## [Decision Letter · Decision Letter 0]

9 Feb 2025

PONE-D-24-56808Calculation method for the volume of weak waterflood zones between layers in heterogeneous oil reservoirs under radial flow conditionsPLOS ONE

Dear Dr. Zhang,

Thank you for submitting your manuscript to PLOS ONE. After careful consideration, we feel that it has merit but does not fully meet PLOS ONE’s publication criteria as it currently stands. Therefore, we invite you to submit a revised version of the manuscript that addresses the points raised during the review process.

The authors are suggested to consider the reviewers' comments. Further, include accuracy of the numerical simulation method in the revised paper.

We look forward to receiving your revised manuscript.

Kind regards,

Muhammad Shakaib, PhD

Academic Editor

PLOS ONE

Journal Requirements:

4. Please ensure that you refer to Figure 1, 6, 7, 8, 9 in your text as, if accepted, production will need this reference to link the reader to the figure.

5. Please upload a copy of Figure 10, 11, 12, 13, to which you refer in your text on page 14, 15, 16, 17. If the figure is no longer to be included as part of the submission please remove all reference to it within the text.

6. We note you have included a table to which you do not refer in the text of your manuscript. Please ensure that you refer to Table 1 in your text; if accepted, production will need this reference to link the reader to the Table.

7. Please include a copy of Table 5 which you refer to in your text on page 17.

Reviewers' comments:

Reviewer's Responses to Questions

**Comments to the Author**

1. Is the manuscript technically sound, and do the data support the conclusions?

Reviewer #1: Yes

Reviewer #2: No

Reviewer #3: Yes

Reviewer #4: Yes

2. Has the statistical analysis been performed appropriately and rigorously? 

Reviewer #1: No

Reviewer #2: No

Reviewer #3: Yes

Reviewer #4: Yes

3. Have the authors made all data underlying the findings in their manuscript fully available?

Reviewer #1: Yes

Reviewer #2: No

Reviewer #3: Yes

Reviewer #4: Yes

4. Is the manuscript presented in an intelligible fashion and written in standard English?

Reviewer #1: No

Reviewer #2: Yes

Reviewer #3: Yes

Reviewer #4: Yes

5. Review Comments to the Author

Reviewer #1: Some comments can improve manuscript:

1. The manuscript offers a novel approach by integrating Darcy's law and Buckley-Leverett theory to predict weak waterflood zones. However, a stronger emphasis on how this method significantly differs from or improves upon existing approaches is needed in the introduction and conclusion.

2. While the iterative calculation method is well described, the step-by-step explanation may be overly detailed for readers with a general reservoir engineering background. Consider moving the derivation of equations to an appendix or supplementary materials and providing a summary of the approach in the main text for better readability.

3. The validation section compares the proposed model with numerical simulations, but the results are limited to a single field case. It would strengthen the manuscript to include additional field data or numerical scenarios to test the robustness of the method across various reservoir conditions.

4. The manuscript mentions a relative error of less than 5% between calculated and simulation results. This is impressive, but a detailed discussion of the factors influencing this error (e.g., heterogeneity, capillary forces, etc.) is needed.

5. Figures 4 and 5 (distribution of residual oil at different stages) are informative but could benefit from clearer labels and descriptions. Indicating specific regions of interest, such as high- and low-permeability zones, would enhance interpretability.

6. Table 1 and Table 2 present essential data on weak waterflood zone proportions but lack sufficient explanation. Brief discussions accompanying these tables in the main text would improve their value to readers.

7. The application section suggests that the method can guide reservoir management strategies. However, more concrete examples of how this method could influence operational decisions, such as well placement or EOR (Enhanced Oil Recovery) techniques, would add value

8. The manuscript highlights the use of a Python program for iterative calculations. It would be helpful to include a brief description of the program's inputs, outputs, and usability, possibly in the supplementary materials.

9. The assumptions made in the model, such as neglecting gravity and capillary pressure, are reasonable for simplification but could limit applicability to certain reservoir conditions. Discussing these limitations explicitly would enhance the manuscript's rigor.

10. The text contains occasional grammatical and stylistic errors that could be addressed to improve readability. For example, phrases such as "offers both high accuracy and computational simplicity" are repeated and could be streamlined for conciseness.

11. Several key references are cited to justify the study. However, recent advancements in reservoir modeling, especially those involving machine learning or advanced numerical methods, are missing. Including such references would improve the context of the study. Some studies can be considered as following: https://doi.org/10.1016/j.geoen.2024.213533

https://doi.org/10.1016/j.watres.2024.122706

https://doi.org/10.46690/ager.2024.11.05

https://doi.org/10.1016/j.eswa.2024.123944

Reviewer #2: 1- Title is very simple and it doe es not show the novelty of the study

2- The parameters introduced, such as permeability contrast, viscosity ratios, and flow resistance coefficients, are physically meaningful and relevant to the study of waterflooding in heterogeneous reservoirs. also there is no physical meaning by adding SW or So in the equation without logic.

The use of dimensionless parameters like the flow resistance coefficient and permeability contrast (e.g., 2.4 times) is logical and aligns with standard practices in reservoir engineering.

However, some parameters (e.g., the threshold permeability contrast of 2.4) should be better justified with physical or experimental reasoning rather than being presented as purely numerical results.

3- Provide clearer explanations for why specific thresholds (e.g., 2.4 times permeability contrast) were chosen and their practical implications in field applications.

Ensure that all derived equations and parameters are dimensionally consistent and validated against real-world data.

4-There is no explicit mention of a sensitivity analysis to evaluate how changes in key parameters (e.g., permeability, viscosity, or injection rates) affect the results. This is a critical omission, as sensitivity analysis is essential for understanding the robustness of the proposed method. Present the results of the sensitivity analysis in graphical or tabular form for better visualization.

5- The paper does not clearly discuss the limitations of the proposed method, such as assumptions about incompressible fluids or neglecting capillary and gravitational effects.

6- While numerical simulations are used for validation, there is no mention of field data validation, which would strengthen the credibility of the method.

7-The grammar is generally acceptable, but there are minor issues with sentence structure and word choice that could be improved for clarity. For example:

"The advancing distance of the waterflooding front in the low-permeability layer reveals that..." could be rephrased for better readability. Some sentences are overly long and could be split into shorter, more concise statements.

8- Proofread the manuscript for grammatical errors and awkward phrasing.

9- Use active voice where possible to improve readability (e.g., "We calculated the advancing distance..." instead of "The advancing distance was calculated...").

10- The paper lacks a discussion on the economic feasibility of the proposed methods (e.g., polymer injection vs. profile modification).

11- Figures and tables are referenced but not included in the text, making it difficult to assess their quality and relevance.

12 also they mentioned table 5 but there is not table 45 in the manuscript, same thing for some figures and not existing such as Figure 11.

13- the discussion part is very shallow

Reviewer #3: 1. Some typo errors in the article should be corrected.

2. It is recommended to add some suggestions for future works in this area to improve the conclusion.

3. The Abstract should contain answers to the following questions: What problem was studied, and why is it important? What methods were used? What are the important results? What conclusions can be drawn from the results? What is the novelty of the work, and where does it go beyond previous efforts in the literature? Please include specific and quantitative results in the Abstract while ensuring that it is suitable for a broad audience. References, figures, tables, equations, and abbreviations should be avoided.

4. The "Introduction" section should be impressive and informative. Some relative papers are highly recommended in field of analytical method to improve this section.

Python approach for using homotopy perturbation method to investigate heat transfer problems, Radiative effects on 2D unsteady MHD Al2O3‐water nanofluid flow between squeezing plates: A comparative study using AGM and HPM in Python, Heat transfer analysis of unsteady nanofluid flow between moving parallel plates with magnetic field: Analytical approach, Thermal analysis of Non-Newtonian visco-inelastic fluid MHD flow between rotating disks, Thermal analysis of boundary layer nanofluid flow over the movable plate with internal heat generation, radiation, and viscous dissipation

5. The mathematical equations must be double-checked.

6. The authors should try to give advantages of using their method compared to others.

7. The work does not provide a well-written conclusion section.

8. There is needed to be more physically discussed.

9. You should provide more information about your solution method.

10. In the conclusion and also in the abstract, authors must present their main outcomes in quantitative outcomes.

Reviewer #4: Manuscript: Calculation method for the volume of weak waterflood zones between layers in heterogeneous oil reservoirs under radial flow conditions.

The following comments are recommended for the manuscript and I think it needs the minor revision at this stage:

1- In this paper the water drive front advancement equations was obtained based on Darcy's Law, please calculate the Reynolds number in flow conditions to check whether Darcy's law is valid or not?

2- Please re-check the values of permeability in lines 300-301 on page 12.

3- Please re-check and correct the figures’ number in line 361 on page 14 to the end of manuscript.

6. PLOS authors have the option to publish the peer review history of their article (what does this mean? ). If published, this will include your full peer review and any attached files.

**Do you want your identity to be public for this peer review?** For information about this choice, including consent withdrawal, please see our Privacy Policy .

Reviewer #1: **Yes: ** Hung Vo Thanh

Reviewer #2: **Yes: ** TAREK AL ARBI GANAT

Reviewer #3: No

Reviewer #4: No

---

## [Author Response · Author response to Decision Letter 1]

18 Mar 2025

Title: Calculation method for the volume of weak waterflood zones between layers in heterogeneous oil reservoirs under radial flow conditions (PONE-D-24-56808)

Dear Editors and Reviewers,

Thank you very much for your valuable comments on our manuscript. We are sorry that the manuscript does not fully meet your requirements and publication quality. Those comments are all valuable and very helpful for revising and improving our manuscript, as well as the important guiding significance to our researches.

Since receiving your letter, we have studied the comments carefully and revised the manuscript point by point. The responses to editor and reviewers’ comments are appended below.

We hope the revised manuscript will meet with your approval and be suitable for publication in PLOS ONE. We would be glad to revise the manuscript further, if necessary.

Thanks and best regards.

Yours sincerely,

Guicai Zhang, PhD

Responses to the comments of editor:

Author reply: We have revised the manuscript and file naming format according to the format requirements of The PLOS ONE. If there are still any issues, we will be happy to make further modifications.

Author reply: The code in this article is only used for simplifying the calculation process and quickly obtaining calculation results. Whether programming code is used or not does not affect the calculation method and results. Therefore, after careful consideration, we have simplified the code section.

Author reply: We have revised the abstract of the manuscript and made it consistent with the abstract in the online submission form.

4. Please ensure that you refer to Figure 1, 6, 7, 8, 9 in your text as, if accepted, production will need this reference to link the reader to the figure.

Author reply: We have checked and revised the figure numbering and reference in the manuscript to ensure that readers can link the content of the manuscript to the figures.

5. Please upload a copy of Figure 10, 11, 12, 13, to which you refer in your text on page 14, 15, 16, 17. If the figure is no longer to be included as part of the submission please remove all reference to it within the text.

Author reply: We have revised the figure numbering and removed the reference to the figures that is no longer to be included as part of the submission.

6. We note you have included a table to which you do not refer in the text of your manuscript. Please ensure that you refer to Table 1 in your text; if accepted, production will need this reference to link the reader to the Table.

Author reply: We have checked and revised the table numbering and reference in the manuscript to ensure that readers can link the content of the manuscript to the tables.

7. Please include a copy of Table 5 which you refer to in your text on page 17.

Author reply: We have revised the table numbering and removed the reference to the tables that is no longer to be included as part of the submission.

Author reply: We have revised the Supporting Information files.

Responses to the comments of reviewer #1:

Thank you very much for your thorough and constructive review. We sincerely appreciate the valuable feedback and suggestions provided by the reviewer.

1. The manuscript offers a novel approach by integrating Darcy's law and Buckley-Leverett theory to predict weak waterflood zones. However, a stronger emphasis on how this method significantly differs from or improves upon existing approaches is needed in the introduction and conclusion.

Author reply: Thank you so much for being able to point out this important issue. I apologize for not expressing myself clearly. Based on your opinions and suggestions, we have further supplemented the progress and shortcomings of existing research in the introduction, pointed out the improvements of the new method, and also supplemented and improved the conclusion. (Page2-3�lines 50-84)

2. While the iterative calculation method is well described, the step-by-step explanation may be overly detailed for readers with a general reservoir engineering background. Consider moving the derivation of equations to an appendix or supplementary materials and providing a summary of the approach in the main text for better readability.

Author reply: Thank you very much for pointing out the problems with this manuscript. We have reduced the description of the iterative calculation method by removing some repetitive derivations, making the derivation process of the calculation method more concise and readable. (Pages 4-11)

3. The validation section compares the proposed model with numerical simulations, but the results are limited to a single field case. It would strengthen the manuscript to include additional field data or numerical scenarios to test the robustness of the method across various reservoir conditions.

Author reply: Thank you for pointing this out and let's supplement the problem. We apologize for not making this clear in the manuscript. We supplemented the validation model by establishing a numerical simulation conceptual model using additional field data, and compared the displacement results of the numerical simulation and calculation method under different parameter conditions to verify the robustness of the calculation method under various reservoir conditions. (Page 16, lines 430-442)

4. The manuscript mentions a relative error of less than 5% between calculated and simulation results. This is impressive, but a detailed discussion of the factors influencing this error (e.g., heterogeneity, capillary forces, etc.) is needed.

Author reply: Thank you very much for pointing out the problems with this manuscript. In response to your comments and suggestions, we have supplemented the possible causes of this error to further improve the content of the manuscript. (Page3-4, lines 107-128; Page 16, lines 417-423)

5. Figures 4 and 5 (distribution of residual oil at different stages) are informative but could benefit from clearer labels and descriptions. Indicating specific regions of interest, such as high- and low-permeability zones, would enhance interpretability.

Author reply: Thank you for highlighting this issue. Based on your suggestion, we have provided more specific annotations and explanations for the content shown in new Figures 6, 7 and 8.

6. Table 1 and Table 2 present essential data on weak waterflood zone proportions but lack sufficient explanation. Brief discussions accompanying these tables in the main text would improve their value to readers.

Author reply: Thank you for pointing out this problem to this manuscript. We apologize for not providing sufficient explanation for the data in Tables. We have supplemented the discussion of all data in the tables of the manuscript, hoping to help readers read and understand.

7. The application section suggests that the method can guide reservoir management strategies. However, more concrete examples of how this method could influence operational decisions, such as well placement or EOR (Enhanced Oil Recovery) techniques, would add value

Author reply: We are sorry that the guidance on reservoir development strategies and methods in the manuscript did not provide specific explanations. Based on your opinions and suggestions, we have provided examples of how calculation methods affect operational decisions. (Page 22, lines 580-590)

8. The manuscript highlights the use of a Python program for iterative calculations. It would be helpful to include a brief description of the program's inputs, outputs, and usability, possibly in the supplementary materials.

Author reply: Thank you for pointing out the problem with the Python program. The code in this article is only used for simplifying the calculation process and quickly obtaining calculation results. Whether programming code is used or not does not affect the prediction method and results. Therefore, after careful consideration, we have simplified the code section.

9. The assumptions made in the model, such as neglecting gravity and capillary pressure, are reasonable for simplification but could limit applicability to certain reservoir conditions. Discussing these limitations explicitly would enhance the manuscript's rigor.

Author reply: Thank you very much for your careful review, let us realize that the discussion of model assumptions is not rigorous enough. Therefore, based on your opinions and suggestions, we have further supplemented the description and discussion of the model hypothesis section. (Page 3, lines 275-284)

10. The text contains occasional grammatical and stylistic errors that could be addressed to improve readability. For example, phrases such as "offers both high accuracy and computational simplicity" are repeated and could be streamlined for conciseness.

Author reply: Thank you very much for pointing out the language error. We have further checked and corrected grammatical and stylistic errors in the manuscript, and simplified phrases with high repetition.

11. Several key references are cited to justify the study. However, recent advancements in reservoir modeling, especially those involving machine learning or advanced numerical methods, are missing. Including such references would improve the context of the study. Some studies can be considered as following: https://doi.org/10.1016/j.geoen.2024.213533

https://doi.org/10.1016/j.watres.2024.122706

https://doi.org/10.46690/ager.2024.11.05

https://doi.org/10.1016/j.eswa.2024.123944

Author reply: Thank you very much for your statement application question. We have organized and supplemented the references, as in this manuscript, the numerical simulation method has been historically fitted and compared with the calculated results of the derived calculation method to verify the accuracy of the method. Therefore, we have not introduced and used a diversified new method for numerical simulation.

We once again thank the reviewer for the insightful comments and suggestions, which have greatly improved the quality of our manuscript. If there are any unreasonable points in the revised manuscript, we will actively make further revisions.

Responses to the comments of reviewer #2:

We greatly appreciate your deep insights and expertise, and we have addressed each of your comments and suggestions individually. Below are our responses to each point.

1. Title is very simple and it doe es not show the novelty of the study

Author reply: Thank you very much for your important suggestions for this manuscript. Based on your feedback and suggestions, we have made some revisions to the title of the manuscript.

2. The parameters introduced, such as permeability contrast, viscosity ratios, and flow resistance coefficients, are physically meaningful and relevant to the study of waterflooding in heterogeneous reservoirs. also there is no physical meaning by adding SW or So in the equation without logic.

The use of dimensionless parameters like the flow resistance coefficient and permeability contrast (e.g., 2.4 times) is logical and aligns with standard practices in reservoir engineering.

However, some parameters (e.g., the threshold permeability contrast of 2.4) should be better justified with physical or experimental reasoning rather than being presented as purely numerical results.

Author reply: Thank you very much for pointing out the issues with the parameters used in this manuscript. In order to illustrate the logic of adding water saturation to the equation, we have provided additional explanations on the source of the equation to improve the logic of the derivation process. For the comparison of threshold permeability, we have analyzed and explained the calculation results based on your opinions and suggestions. (Page 8, lines 228-231 and Page 19-20, lines 507-524)

3. Provide clearer explanations for why specific thresholds (e.g., 2.4 times permeability contrast) were chosen and their practical implications in field applications.

Ensure that all derived equations and parameters are dimensionally consistent and validated against real-world data.

Author reply: Many thanks for such important guidance. It was indeed an oversight on our part not to conduct detailed analysis to the data. In fact, we did not choose 2.4 times permeability contrast as a specific threshold, 2.4 times permeability contrast was the critical value for reducing the volume fraction of the weak water drive zone to 0. When the permeability difference is greater than 2.4, the calculated volume fraction of the weak water drive zone is greater than 0 (Page 19, lines 498-501). We have provided an example in the article to demonstrate the calculation process of a set of data for the derived calculation equation, and the calculation results are consistent with the model in terms of numerical and scale (Page 13-14, lines 346-385).

4. There is no explicit mention of a sensitivity analysis to evaluate how changes in key parameters (e.g., permeability, viscosity, or injection rates) affect the results. This is a critical omission, as sensitivity analysis is essential for understanding the robustness of the proposed method. Present the results of the sensitivity analysis in graphical or tabular form for better visualization.

Author reply: Thank you very much for pointing out the issue regarding sensitivity analysis. Based on your feedback and suggestions, we have supplemented the model validation section by controlling variables and analyzing the calculation results of the volume proportion of weak water drive zones under different key parameters to ensure the applicability of the new predictive calculation model under different conditions. (Page 16, lines 424-434)

5. The paper does not clearly discuss the limitations of the proposed method, such as assumptions about incompressible fluids or neglecting capillary and gravitational effects.

Author reply: Thank you for pointing out the imperfections in the theoretical assumptions.

Regarding the assumption that the oil and water phases are Darcy linear flow, experimental and oilfield data show that due to the low fluid flow velocity in the reservoir, the fluid flow law generally follows Darcy's flow law. Linear flow is one of the typical flow modes in actual reservoir production, and the flow between injection wells and production wells in many reservoirs can be regarded as planar linear flow. Therefore, exploring the flow law under linear flow conditions is also of great research significance.

Regarding the assumption that the effects of gravity, capillary pressure, and start-up pressure gradient are not considered, and that rocks and fluids are incompressible, due to the main exploration of the water drive law in oil fields, according to the study of oil and gas reservoir permeability mechanics, in water injection development oil fields, if the injection of water has a strong effect, the elastic effect of the liquid and rock can usually be ignored. At this time, rocks and liquids can be regarded as rigid medi

---

## [Decision Letter · Decision Letter 1]

4 Apr 2025

New calculation model and application research on weak water-flushed zones distribution prediction in radial flow well patterns of heterogeneous oil reservoirs

PONE-D-24-56808R1

Dear Dr. Zhang,

We’re pleased to inform you that your manuscript has been judged scientifically suitable for publication and will be formally accepted for publication once it meets all outstanding technical requirements.

Kind regards,

Muhammad Shakaib, PhD

Academic Editor

PLOS ONE

Additional Editor Comments (optional):

Reviewers' comments:

Reviewer's Responses to Questions

**Comments to the Author**

1. If the authors have adequately addressed your comments raised in a previous round of review and you feel that this manuscript is now acceptable for publication, you may indicate that here to bypass the “Comments to the Author” section, enter your conflict of interest statement in the “Confidential to Editor” section, and submit your "Accept" recommendation.

Reviewer #1: All comments have been addressed

Reviewer #3: (No Response)

Reviewer #4: All comments have been addressed

2. Is the manuscript technically sound, and do the data support the conclusions?

Reviewer #1: Yes

Reviewer #3: (No Response)

Reviewer #4: Yes

3. Has the statistical analysis been performed appropriately and rigorously? 

Reviewer #1: Yes

Reviewer #3: (No Response)

Reviewer #4: Yes

4. Have the authors made all data underlying the findings in their manuscript fully available?

Reviewer #1: Yes

Reviewer #3: (No Response)

Reviewer #4: Yes

5. Is the manuscript presented in an intelligible fashion and written in standard English?

Reviewer #1: Yes

Reviewer #3: (No Response)

Reviewer #4: Yes

6. Review Comments to the Author

Reviewer #1: (No Response)

Reviewer #3: The authors have responded to my previous comments and revised the manuscript accordingly. I believe it is acceptable for publication in its form.

Reviewer #4: I believe the authors have adequately addressed the issues, and I recommend publishing the manuscript in its current form in the PLOS ONE.

7. PLOS authors have the option to publish the peer review history of their article (what does this mean? ). If published, this will include your full peer review and any attached files.

**Do you want your identity to be public for this peer review?** For information about this choice, including consent withdrawal, please see our Privacy Policy .

Reviewer #1: No

Reviewer #3: No

Reviewer #4: No

---

## [Editor Report · Acceptance letter]

PONE-D-24-56808R1

PLOS ONE

Dear Dr. Zhang,

I'm pleased to inform you that your manuscript has been deemed suitable for publication in PLOS ONE. Congratulations! Your manuscript is now being handed over to our production team.

Kind regards,

on behalf of

Dr. Muhammad Shakaib

Academic Editor

PLOS ONE